# Variance as a Catalyst: Efficient and Transferable Semantic Erasure Adversarial Attack for Customized Diffusion Models

Jiachen Yang [1]   Yusong Wang [1]   Yanmei Fang [1 2]   Yunshu Dai [1]   Fangjun Huang [1 2]

## Abstract

Latent Diffusion Models (LDMs) enable fine-tuning with only a few images and have become widely used on the Internet. However, it can also be misused to generate fake images, leading to privacy violations and social risks. Existing adversarial attack methods primarily introduce noise distortions to generated images but fail to completely erase identity semantics. In this work, we identify the variance of VAE latent code as a key factor that influences image distortion. Specifically, larger variances result in stronger distortions and ultimately erase semantic information. Based on this finding, we propose a Laplace-based (LA) loss function that optimizes along the fastest variance growth direction, ensuring each optimization step is locally optimal. Additionally, we analyze the limitations of existing methods and reveal that their loss functions often fail to align gradient signs with the direction of variance growth. They also struggle to ensure efficient optimization under different variance distributions. To address these issues, we further propose a novel Lagrange Entropy-based (LE) loss function. Experimental results demonstrate that our methods achieve state-of-the-art performance on CelebA-HQ and VGGFace2. Both proposed loss functions effectively lead diffusion models to generate pure-noise images with identity semantics completely erased. Furthermore, our methods exhibit strong transferability across diverse models and efficiently complete attacks with minimal computational resources. Our work provides a practical and efficient solution for privacy protection.

[1]School of Cyber Science and Technology, Shenzhen Campus of Sun Yat-sen University, Shenzhen 518106, China. [2]Guangdong Provincial Key Laboratory of Information Security Technology, Guangzhou 510006, China. Correspondence to: Yanmei Fang <fangym@mail.sysu.edu.cn>.

*Proceedings of the 42nd International Conference on Machine Learning*, Vancouver, Canada, PMLR 267, 2025. Copyright 2025 by the author(s).

## 1. Introduction

Latent Diffusion Models (LDMs) have revolutionized the field of generative models (Song & Ermon, 2019; Ho et al., 2020; Nichol & Dhariwal, 2021). They enable generate high-quality and diverse images that closely resemble human-generated content (Rombach et al., 2022; Peebles & Xie, 2023; Podell et al., 2024; Labs, 2023; Esser et al., 2024). This breakthrough in generative modeling is largely attributed to the ability of LDMs to perform few-shot fine-tuning (Gal et al., 2022; Hu et al., 2021; Ruiz et al., 2023; Kumari et al., 2023; Ye et al., 2023). With as few as 4 to 5 images, LDMs can learn new concepts, such as specific human faces or unique artistic styles. This capability has enabled their widespread use in academia and industry, driving advancements in image synthesis, personalization, and artistic creation. However, this powerful capability has also raised significant privacy concerns. Malicious users can exploit LDMs to generate fake images using a small set of personal data, such as photos from social media. This poses serious risks to individual privacy and reputation (Higgins, 2023). These risks underscore the urgent need for mitigation strategies to address this growing issue.

Under these challenges, researchers have proposed adversarial attacks on diffusion models to prevent them from learning data features. These attacks can be divided into three categories: maximizing the training loss of UNet (Liang et al., 2023; Van Le et al., 2023; Xue et al., 2023; Wang et al., 2024; Liu et al., 2024b), targeting the cross-attention module within UNet (Xu et al., 2024; Liu et al., 2024a; Lo et al., 2024), and attacking the VAE encoder (Salman et al., 2023; Shan et al., 2023; Liang & Wu, 2023; Li et al., 2024). The first two types often rely on specific prompts and model architectures when designing perturbations, which limits their transferability. Furthermore, due to the large number of parameters in UNet, these methods are computationally expensive and time-consuming. These limitations make them impractical for rapidly evolving models or resource-constrained scenarios. In contrast, VAE-based attacks are generally prompt-independent since the VAE does not participate in encoding prompts during image generation. However, these methods mainly focus on maximizing the distribution differences of latent codes between clean

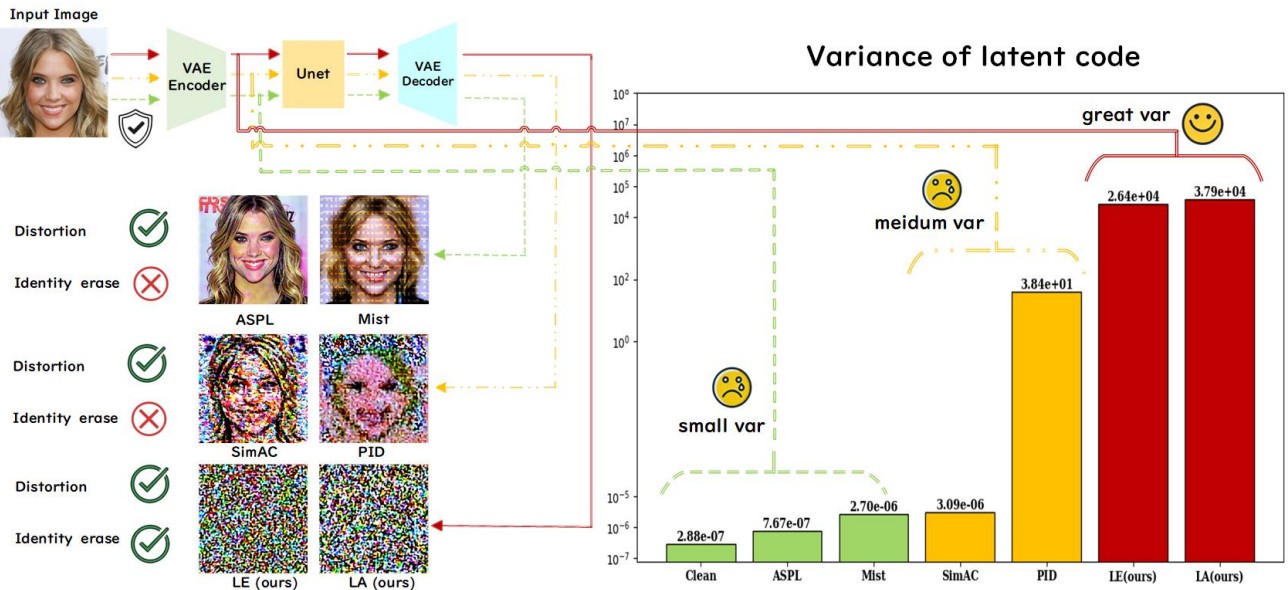

*Figure 1.* We find that larger variances in the VAE latent code improve the effectiveness of protection methods. Smaller variances, as in ASPL and Mist, still preserve identity semantics, while larger variances, as in SimAC and PID, remove most of them. Our proposed Laplace Loss (LA) and Lagrange Entropy Loss (LE) achieve even higher variances, generating pure-noise images with semantics completely erased.

and adversarial images or optimizing their mean and variance. As a result, they only introduce varying levels of noise to the generated images but fail to effectively remove facial semantic information.

To address the aforementioned issues, we perform an in-depth investigation of the VAE encoder. PID (Li et al., 2024) observes that the mean of the VAE latent code influences the structure and texture of the generated image, while the variance governs the model's ability to capture core concepts and affects semantic diversity. They suggest that targeting both mean and variance is essential for achieving optimal attack performance. However, through a further analysis of VAE properties, we find that **variance is the key factor determining the level of distortion and identity semantic erasure in generated images**. As shown in Figure 1, we encode images protected by various methods into their respective latent codes and analyze their variances. For methods causing minimal distortion, such as ASPL and Mist, the variances of their latent codes closely resemble that of the clean image, thereby preserving most facial semantics. In contrast, methods with medium variances, such as SimAC and PID, introduce significant distortion but still fail to completely erase identity semantics.

Building on this observation, we investigate how the perturbation matrix in the pixel space influences the variance matrix in the latent space through the VAE encoder. We find that the direction of variance growth is closely related to

the gradient sign of the loss function with respect to variance. Existing methods like Mist (Liang & Wu, 2023) uses Mean Squared Error (MSE) to maximize the variance gap between clean images and adversarial examples. However, this approach misaligns the gradient sign with the direction of variance growth, causing local optimization traps in the early stages and hindering effective variance increase. PID (Li et al., 2024) addresses this issue by applying a logarithmic transformation to the variance, ensuring better alignment between the gradient direction and variance growth. Despite this improvement, PID faces diminishing optimization space as variance becomes more uniform, leading to slow optimization. In its original work, PID requires 900 steps to remove most identity semantics. It lacks an in-depth theoretical analysis to uncover the root cause of this issue, leaving the problem unexplored. To address these limitations, we propose two novel loss functions: Laplace Loss (LA) and Lagrange-Entropy Loss (LE). LA consistently optimizes toward increasing variance regardless of its distribution, making each update locally optimal. LE combines entropy and Lagrange terms to balance optimization, ensuring alignment with variance growth while maintaining sufficient optimization space. Both methods can completely erase identity semantics and generate pure noise images in 30 steps, achieving a 30× speedup over PID. Our main contributions are summarized as follows:

- We observe that the variance of latent codes directly

affects the semantic integrity of generated images. Furthermore, we find that the optimization direction of the variance is closely tied to the gradient sign of the loss function with respect to the variance.

- We analyze prior methods and identify that their gradient signs often misalign with the variance growth or that the optimization space of variance becomes progressively compressed. To overcome these issues, we propose two novel loss functions: LA and LE.

- Our methods achieve state-of-the-art performance on two facial image datasets. They not only generate pure-noise images with completely erased identity semantics, but also offer better transferability, lower computational cost, and faster attack speeds.

## 2. Related Work

### 2.1. Diffusion Models

Diffusion models consist of two main phases: the diffusion process and the reverse diffusion process. Given an image $x_0$, the diffusion process progressively adds noise to the data distribution based on a predefined noise schedule $\{\beta_t\}_{t=1}^T$. As noise accumulates, the image $x_T$ becomes increasingly noisy. When $T$ is sufficiently large, the image ultimately transforms into pure Gaussian noise. This process is formally described as follows:

$$x_t = \sqrt{1-\beta_t}x_{t-1} + \sqrt{\beta_t}\epsilon_t = \sqrt{\bar{\alpha}_t}x_0 + \sqrt{1-\bar{\alpha}_t}\epsilon \tag{1}$$

where $\alpha_t = 1 - \beta_t$, $\bar{\alpha}_t = \prod_{i=1}^t \alpha_i$ and $\epsilon \sim \mathcal{N}(0, I)$ is the noise sampled from a normal distribution.

The reverse process aims to predict the noise $\epsilon_t$ added at the time step $t$ with a denoising model $\epsilon_\theta(x_{t+1}, t)$. Through this multi-step denoising process, the model gradually recovers the input image $x_0$. Consequently, the training goal of the reverse process is to minimize the error between the estimated noise with prompt $c$ and the true noise.

$$\mathcal{L}_{cond}(\theta, x_0) = \mathbb{E}_{x_0,t,c,\epsilon \sim \mathcal{N}(0,1)} \|\epsilon - \epsilon_\theta(x_{t+1}, t, c)\|_2^2 \tag{2}$$

### 2.2. Personalized Systhesis

DreamBooth (Ruiz et al., 2023) personalizes text-to-image models for a target concept by learning from a few reference images with a customized prompt like *"a photo of sks person"*, where *"sks"* represents the learned concept. It uses a generic prompt $c_p$, *"a photo of person"*, and a prior preservation loss with a hyperparameter $\lambda$ to balance personalization and generic information, addressing overfitting and text-shifting. The training combines two objectives:

$$\mathcal{L}_{db}(\theta, x_0) = \mathbb{E}_{x_0,t,t',\epsilon,\epsilon' \sim \mathcal{N}(0,1)} \|\epsilon - \epsilon_\theta(x_{t+1}, t, c)\|_2^2$$
$$+ \lambda \|\epsilon' - \epsilon_\theta(x'_{t+1}, t', c_p)\|_2^2 \tag{3}$$

In addition to DreamBooth, lightweight personalization techniques such as Textual Inversion (Gal et al., 2022), LoRA (Hu et al., 2021), and IP-Adapter (Ye et al., 2023) have gained popularity. Notably, LoRA is widely adopted in online communities for creating character portraits and imitating artworks. However, its use raises significant privacy concerns and art copyright infringement issues.

### 2.3. Adversarial Attacks for Diffusion Models

Traditional adversarial attacks primarily target classification models (Goodfellow et al., 2015; Madry et al., 2018). To mitigate the risks of misuse in personalized generation, recent studies propose adversarial attacks to prevent LDMs from learning unauthorized images. These works can be mainly divided into three categories. The first category aims to maximize the training loss of diffusion model and force the predicted noise to deviate from the real noise (Liang et al., 2023; Van Le et al., 2023; Liang & Wu, 2023; Xue et al., 2023; Wang et al., 2024; Liu et al., 2024b). The second manipulates cross-attention in UNet to weaken the weight of identity keywords in prompts, such as *"sks"* in the prompt *"a photo of sks person,"* in the cross-attention map. This reduces the connection between the prompt and pixel distribution of the generated images (Xu et al., 2024; Liu et al., 2024a; Lo et al., 2024). However, both approaches depend on UNet gradients and specific prompts, leading to poor transferability and high computational costs. The third targets the VAE module, which operates independently of the diffusion model. These methods either maximize the divergence between the latent distributions of clean images and adversarial examples (Salman et al., 2023; Shan et al., 2023; Liang & Wu, 2023) or jointly attack the latent mean and variance (Li et al., 2024). Although these methods can generate distorted images with varying levels of noise, they fail to completely erase identity semantic features.

## 3. Method

### 3.1. In-deep Analysis of VAE Encoder

Adversarial attacks for diffusion models predominantly rely on Projected Gradient Descent (PGD) (Madry et al., 2018) to iteratively update the perturbations. PGD optimizes the perturbation $\delta$ added to the input $x$, aiming to maximize a loss function $\mathcal{L}$ under the constraint $\|\delta\|_p \leq \epsilon$. The PGD update rule at step $t$ with step size $\alpha$ is given as:

$$\delta^{t+1} = \Pi_{\mathbb{B}_\epsilon}\left(\delta^t + \alpha \cdot \text{sign}\left(\frac{\partial \mathcal{L}}{\partial \delta}\right)\right), \tag{4}$$

In PGD, $\text{sign}\left(\frac{\partial \mathcal{L}}{\partial \delta}\right)$ determines the update direction of perturbation $\delta$. Therefore, when analyzing the effect of perturbation $\delta$ on the variance $\sigma^2$, we decompose $\frac{\partial \mathcal{L}}{\partial \delta}$ using the

chain rule as follows:

$$\frac{\partial \mathcal{L}}{\partial \delta} = \frac{\partial \mathcal{L}}{\partial \sigma^2} \cdot \frac{\partial \sigma^2}{\partial \delta} \tag{5}$$

The term $\frac{\partial \sigma^2}{\partial \delta}$ directly reflects the relationship between the perturbation matrix $\delta$ and the change in variance matrix $\sigma^2$. Aligning with its sign direction offers the most straightforward path to increasing variance. Then, we further decompose this term into the following chain form with the hidden state $\phi = f(x') = f(x + \delta)$ of VAE encoder hidden layer:

$$\frac{\partial \mathcal{L}}{\partial \delta} = \frac{\partial \mathcal{L}}{\partial \sigma^2} \cdot \frac{\partial \sigma^2}{\partial \delta} = \frac{\partial \mathcal{L}}{\partial \sigma^2} \cdot \frac{\partial \sigma^2}{\partial \phi} \cdot \frac{\partial \phi}{\partial \delta} \tag{6}$$

The VAE encoder is nonlinear, making it difficult to directly analyze the effect of $\delta$ on the hidden state $\phi$. However, when $\delta$ is small, the relationship can be locally approximated as linear. Specifically, for small perturbations $\|\delta\|_p \leq \epsilon$, the first-order Taylor expansion of the VAE encoder $f(x)$ at $x$ provides:

$$\phi = f(x') = f(x + \delta) \approx f(x) + J_f \cdot \delta, \tag{7}$$

The Jacobian matrix $J_f$ of $f$ with respect to the input $x$ captures the first-order partial derivatives of each hidden state $\phi_k$ with respect to each input pixel $\delta_i$. Intuitively, $J_f$ provides a local linear mapping, describing how small changes on perturbations $\delta$ in image space propagate to the hidden state $\phi$ in latent space. The product $J_f \cdot \delta$ represents the locally linearized effect of perturbations $\delta$ on each feature component of the hidden state matrix $\phi$. Given the small magnitude of $\delta$, its effect on the hidden state in latent space can be approximated as:

$$\frac{\partial \phi_k}{\partial \delta_i} \approx J_{f_{(i,k)}}^\top \tag{8}$$

The Jacobian matrix $J_{f_{(i,k)}}$ represents the mapping relationship between the $i$-th perturbation component in the perturbation matrix $\delta$ and the $k$-th hidden state component in the hidden state matrix $\phi$. Similarly, the mapping from the hidden state matrix $\phi$ to the variance matrix $\sigma^2$ is governed by another neural network. This mapping can be linearized using the Jacobian matrix $J_g$, which quantifies how changes in the $k$-th component of $\phi$ affect the $j$-th variance component $\sigma_j^2$. The Jacobian matrix $J_g$ characterizes this relationship as follows:

$$\frac{\partial \sigma_j^2}{\partial \phi_k} \approx J_{g_{(k,j)}}^\top \tag{9}$$

where $J_{g_{(k,j)}}$ quantifies the influence of the $k$-th hidden state $\phi_k$ on the $j$-th variance component $\sigma_j^2$ in the latent space. Consequently, $\frac{\partial \sigma^2}{\partial \delta}$ can be approximated as the product of two Jacobian matrices:

$$M_{i,j} = \frac{\partial \sigma_j^2}{\partial \delta_i} = \sum_k J_{g_{(k,j)}}^\top \cdot J_{f_{(i,k)}}^\top \tag{10}$$

where $M_{i,j}$ represents the influence of the $i$-th perturbation component in $\delta$ on the $j$-th variance component in $\sigma^2$. Using the chain rule, the gradient of the loss $\mathcal{L}$ with respect to the $i$-th perturbation $\delta_i$ can be expressed as:

$$\frac{\partial \mathcal{L}}{\partial \delta_i} = \sum_j \frac{\partial \mathcal{L}}{\partial \sigma_j^2} \cdot \frac{\partial \sigma_j^2}{\partial \delta_i} = \sum_j \frac{\partial \mathcal{L}}{\partial \sigma_j^2} \cdot M_{i,j} \tag{11}$$

Building upon Equation 4, we observe that the update direction of the $i$-th perturbation component in the perturbation matrix is determined by sign $\left(\frac{\partial \mathcal{L}}{\partial \delta}\right)$ and the linear combination of two matrices $M_{i,j}$.

### 3.2. Laplace Approximation is Local Optimum

From Equations 5 and 10, we observe that the sign of $\frac{\partial \sigma_j^2}{\partial \delta_i}$ represents the direction in which the $i$-th perturbation directly affects the increase of the $j$-th variance component. This allows the optimization process to be simplified by assigning a constant gradient to $\frac{\partial \mathcal{L}}{\partial \sigma_j^2}$, ensuring a direct relationship with variance growth. Based on this insight, we propose a Laplace-based (LA) loss function defined as:

$$\mathcal{L}_{Laplace} = \frac{|\sigma^2 - \mu|}{b}, \tag{12}$$

where $\mu$ is the target mean (usually 0), and $b$ is a scaling factor (usually 1). The gradient of this loss function with respect to $\sigma^2$ is a constant value $\frac{1}{b}$. This constant gradient ensures that the optimization direction is solely determined by $\frac{\partial \sigma_j^2}{\partial \delta_i}$, which reflects the inherent relationship between the perturbation and the variance growth direction. Specifically, $\frac{1}{b}$ uniformly scales the gradient across all dimensions, preserving the alignment of optimization with variance growth. Consequently, the LA loss ensures consistent alignment with the variance growth direction, enabling each PGD step to achieve a locally optimal update.

### 3.3. Limitations of Previous Work

Adversarial attacks for variance manipulation rely on loss function, which dictates the gradient direction $\frac{\partial \mathcal{L}}{\partial \delta}$ and directly impacts optimization. In this section, we analyze two common methods for variance optimization: Mean Squared Error (MSE) (Liang & Wu, 2023) and PID (Li et al., 2024), revealing their limitations. The MSE loss is defined as:

$$\mathcal{L}_{MSE} = \sum_j \left(\sigma_j^2 - \sigma_{clean}^2\right)^2 \tag{13}$$

where $\sigma_j^2$ is the $j$-th variance component in the latent space, and $\sigma_{\text{clean}}^2$ represents the variance of clean latent distributions. Its gradient with respect to $\sigma_j^2$ is:

$$\frac{\partial \mathcal{L}_{MSE}}{\partial \sigma_j^2} = 2\left(\sigma_j^2 - \sigma_{clean}^2\right) \tag{14}$$

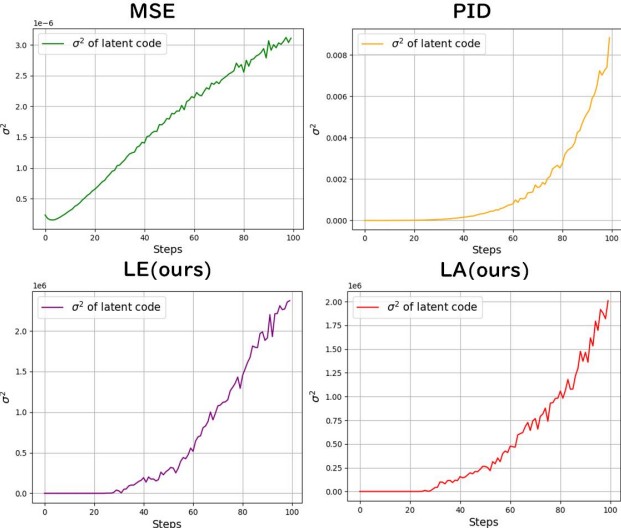

*Figure 2.* Variance growth trends of latent code within 100 steps. MSE is trapped in a local optimum, with variance remaining at the $1e-6$ level. PID increases variance but at a very slow rate. In contrast, our methods, LA and LE, rapidly amplify variance to the $1e5$ level within just 30 to 50 steps.

MSE loss maximizes the squared distance between $\sigma_j^2$ and $\sigma_{clean}^2$. However, it suffers from notable limitations. When variance components are perturbed, the initial variance distribution becomes uneven. In the early stages of optimization, MSE mistakenly prioritizes the few large variance components while neglecting the majority of small variance components. This behavior causes MSE to prioritize large variances, disrupting the optimization process. Specifically, the product of $\frac{\partial \mathcal{L}_{MSE}}{\partial \sigma_j^2}$ and $M_{i,j}$ can result in a negative value for certain perturbations. This misalignment reverses the gradient direction, leading to ineffective variance growth. This issue can be formally expressed as:

$$\text{sign}(\frac{\partial \mathcal{L}_{MSE}}{\partial \delta_i}) = \text{sign}(\sum_j \frac{\partial \mathcal{L}_{MSE}}{\partial \sigma_j^2} \cdot M_{i,j}) \neq \text{sign}(\sum_j M_{i,j})$$

(15)

As shown in Figure 2, at an optimization step of 100, MSE only increases the variance to $3 \times 10^{-6}$, which remains nearly identical to $\sigma_{clean}^2$. This indicates that MSE falls into a local trap at the start of optimization. It fails to effectively amplify the larger variance components it targets and neglects to expand the smaller variance components that require growth. For PID, the loss of variance and gradient are defined as:

$$\mathcal{L}_{PID} = \sum_j \left( \log \sigma_j^2 - \log \sigma_{clean}^2 \right)^2,$$
$$\frac{\partial \mathcal{L}_{PID}}{\partial \sigma_j^2} = \frac{2}{\sigma_j^2} \left( \log \sigma_j^2 - \log \sigma_{clean}^2 \right).$$

(16)

PID prioritizes the optimization of small variance compo-

nents due to its larger gradient values at lower variances. This ensures that the gradient sign $\frac{\partial \mathcal{L}_{PID}}{\partial \sigma_j^2}$ aligns with the variance growth direction $M_{i,j}$. However, PID is less responsive to large variance components. Once small variances are optimized, the variance distribution becomes more uniform, which reduces the optimization space and leads to slow convergence. As illustrated in Figure 2, although PID effectively increases variance at step 100, its growth rate is significantly slower compared to our proposed LA Loss.

### 3.4. Lagrange Entropy Loss

To overcome the limitations of MSE and PID, we propose the Lagrange Entropy (LE) Loss, which integrates an entropy term and a Lagrange constraint to balance variance optimization:

$$\mathcal{L}_{LE} = -\sum_j \sigma_j^2 \log(\sigma_j^2) + \lambda \left( \sum_j \sigma_j^2 - c \right)^2$$

(17)

The entropy term promotes the growth of small variance components and the Lagrange constraint ensures overall variance remains balanced. $\lambda = 0.1$ controls the trade-off between the two terms, and $c = 1$ sets the target variance. The gradient of $L_{LE}$ with respect to $\sigma^2$ is expressed as:

$$\frac{\partial \mathcal{L}_{LE}}{\partial \sigma_j^2} = - \left( \log(\sigma_j^2) + 1 \right) + 2\lambda \left( \sum_j \sigma_j^2 - c \right)$$

(18)

This gradient ensures that small variance components are optimized more strongly at the beginning, driven by the entropy term. As the distribution grows, the Lagrange term maintains sufficient optimization space for larger variances. As shown in Figure 2, LE loss enables efficient variance growth. Both LE and LA losses can completely remove identity semantics within 30–50 steps, significantly outperforming the 900-step requirement of PID.

## 4. Experiments

### 4.1. Experimental Setup

**Datasets:** We compare LA and LE with several representative methods on the CelebA-HQ (Karras et al., 2018) and VGGFace2 (Cao et al., 2018) datasets. We select 50 different identities from each dataset and use 4 reference images per identity as training data. We apply the PGD (Madry et al., 2018) to update the perturbations, setting the perturbation budget $\eta$ to 0.05 and the step size $\alpha$ to 1/255 for all methods. Baseline methods use default settings, and Stable Diffusion v1.5 serves as the base model.

**Evaluation Metrics:** Our goal is to prevent diffusion models from learning core concepts of training images, generating pure noise with erased facial semantics. We generate 100

Table 1. Comparing the performance of our method with baselines against DreamBooth (Ruiz et al., 2023) on CelebA-HQ and VGGFace2. The best result under each metric is marked with **bold**. The prompt used is *"a photo of a sks person."*

| Method | CelebA-HQ | | | | VGGFace2 | | | |
|---|---|---|---|---|---|---|---|---|
| | ISM ↓ | FDFR ↑ | Brisque ↑ | LPIPS ↑ | ISM ↓ | FDFR ↑ | Brisque ↑ | LPIPS ↑ |
| No Defense | 0.608 | 0.041 | 17.896 | 0.662 | 0.638 | 0.025 | 18.193 | 0.724 |
| AdvDM (Liang et al., 2023) | 0.424 | 0.307 | 24.215 | 0.798 | 0.142 | 0.944 | 47.862 | 0.868 |
| ASPL (Van Le et al., 2023) | 0.406 | 0.287 | 24.419 | 0.805 | 0.158 | 0.906 | 46.142 | 0.865 |
| Mist (Liang & Wu, 2023) | 0.249 | 0.169 | 13.981 | 0.707 | 0.246 | 0.257 | 18.324 | 0.756 |
| MetaCloak (Liu et al., 2024b) | 0.593 | 0.051 | 36.325 | 0.712 | 0.525 | 0.059 | 36.771 | 0.747 |
| SimAC (Wang et al., 2024) | 0.253 | 0.865 | 51.059 | 0.823 | 0.196 | 0.981 | 51.874 | 0.836 |
| DisDiff (Liu et al., 2024a) | 0.605 | 0.116 | 29.361 | 0.695 | 0.263 | 0.902 | 43.623 | 0.758 |
| SDS- (Xue et al., 2023) | 0.655 | 0.005 | 38.519 | 0.743 | 0.591 | 0.002 | 37.325 | 0.781 |
| PID (Li et al., 2024) | 0.069 | 0.938 | 85.533 | 0.899 | 0.046 | 0.968 | 86.946 | 0.945 |
| LE(ours) | **0** | **1** | **155.804** | **1.021** | **0** | **1** | **154.494** | **1.028** |
| LA(ours) | **0** | **1** | **155.845** | **0.959** | **0** | **1** | **155.845** | **1.031** |

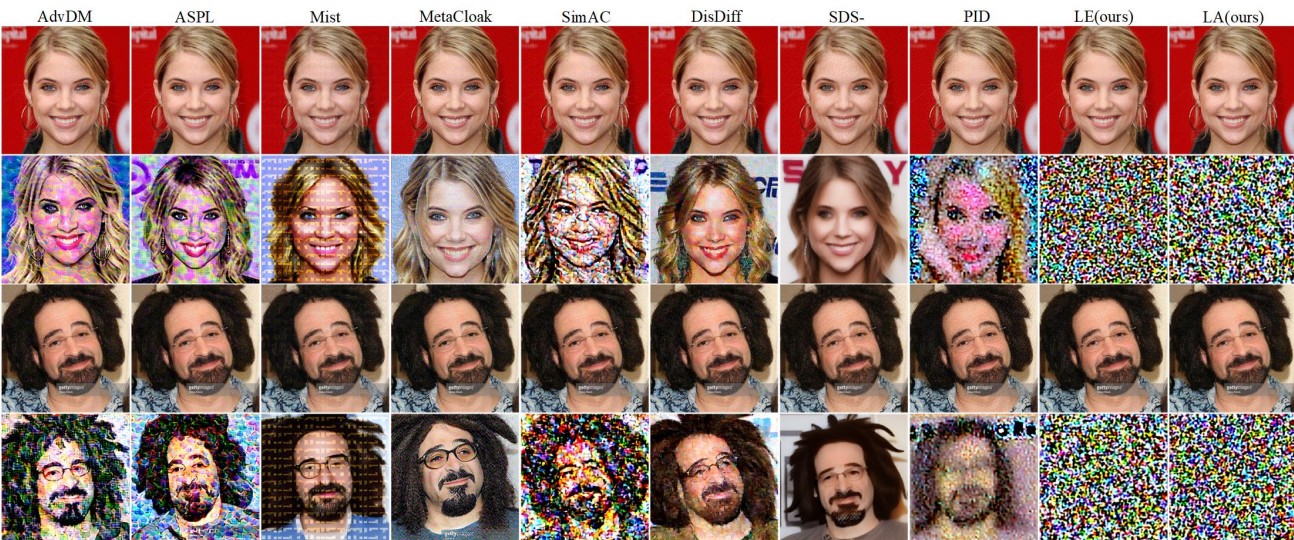

Figure 3. Visualization of our methods and baselines against DreamBooth on CelebA-HQ and VGGFace2. The first and third rows show reference images, while the second and fourth rows display generated images from fine-tuned models.

images for each fine-tuned model and evaluate them using four metrics. ***Face Detection Failure Rate (FDFR)*** measures whether the RetinaFace detector (Deng et al., 2020) recognizes the images as faces; higher FDFR indicates better protection. For detected faces, ***Identity Score Matching (ISM)*** calculates the distance between facial embeddings of generated and reference images; smaller ISM means weaker identity correlation. ***Brisque*** measures image naturalness; higher scores indicate greater deviation and distortion. ***LPIPS*** quantifies perceptual differences; higher scores reflect stronger semantic erasure.

### 4.2. Main Results

**Results against DreamBooth (Ruiz et al., 2023):** Table 1 and Figure 3 show that our methods outperform baselines

on CelebA-HQ and VGGFace2. They achieve ISM of 0 and FDFR of 1 on both datasets, completely removing identity semantics. Figure 3 further demonstrates that our methods generate pure noise images, while baselines retain partial semantics. Higher Brisque and LPIPS scores (155.804 and 1.021) confirm significant image degradation, making the images highly unnatural and unrecognizable.

**Cross-model Transferability against LoRA (Hu et al., 2021):** To evaluate the transferability of our methods across different diffusion model architectures, we use adversarial examples generated on SD1.5 as training data for other models. We test two similar models (SD2.1 and SDXL) and two with significantly different architectures (SD3.5 and FLUX.1-dev). Tables 2 and 3 demonstrate that our methods achieve superior transferability across diverse dif-

*Table 2.* Comparing the transferability of our method with other approaches against **LoRA** (Hu et al., 2021) on CelebA-HQ. The best result under each metric is marked with **bold**.

| Method | SD2.1 | | | | SDXL | | | |
|---|---|---|---|---|---|---|---|---|
| | ISM ↓ | FDFR ↑ | Brisque ↑ | LPIPS ↑ | ISM ↓ | FDFR ↑ | Brisque ↑ | LPIPS ↑ |
| No Defense | 0.729 | 0.073 | 16.409 | 0.669 | 0.791 | 0.001 | 13.483 | 0.515 |
| AdvDM (Liang et al., 2023) | 0.532 | 0.313 | 39.369 | 0.704 | 0.765 | 0.019 | 18.419 | 0.539 |
| ASPL (Van Le et al., 2023) | 0.519 | 0.331 | 39.226 | 0.709 | 0.766 | 0.002 | 20.589 | 0.524 |
| Mist (Liang & Wu, 2023) | 0.179 | 0.231 | 18.097 | 0.677 | 0.583 | 0.126 | 24.143 | 0.622 |
| MetaCloak (Liu et al., 2024b) | 0.635 | 0.087 | 41.381 | 0.699 | 0.738 | 0.002 | 22.766 | 0.517 |
| SimAC (Wang et al., 2024) | 0.401 | 0.642 | 41.409 | 0.733 | 0.746 | 0.018 | 10.459 | 0.546 |
| DisDiff (Liu et al., 2024a) | 0.627 | 0.166 | 40.127 | 0.709 | 0.782 | 0 | 17.366 | 0.519 |
| SDS- (Xue et al., 2023) | 0.673 | 0.016 | 52.108 | 0.711 | 0.732 | 0 | 8.103 | 0.569 |
| PID (Li et al., 2024) | 0.089 | 0.887 | 91.461 | 0.949 | 0.602 | 0 | 17.848 | 0.545 |
| LE(ours) | **0** | **1** | **151.089** | **0.947** | **0.171** | **0.649** | **126.369** | **0.893** |
| LA(ours) | **0** | **1** | **154.724** | **0.955** | **0.178** | **0.401** | **102.815** | **0.822** |

*Table 3.* Comparing the transferability of our method with other approaches against **LoRA** (Hu et al., 2021) on CelebA-HQ. SD3.5 and FLUX.1-dev uses a more advanced text encoder and replaces the UNet with MM-DiT as the denoising network. The best performance under each metric is marked with **bold**.

| Method | SD3.5 | | | | FLUX.1-dev | | | |
|---|---|---|---|---|---|---|---|---|
| | ISM ↓ | FDFR ↑ | Brisque ↑ | LPIPS ↑ | ISM ↓ | FDFR ↑ | Brisque ↑ | LPIPS ↑ |
| No Defense | 0.587 | 0.001 | 0.699 | 0.589 | 0.705 | 0.001 | 7.722 | 0.598 |
| AdvDM (Liang et al., 2023) | 0.532 | 0.002 | 11.755 | 0.612 | 0.739 | 0.009 | 13.268 | 0.651 |
| ASPL (Van Le et al., 2023) | 0.543 | 0.001 | 11.541 | 0.621 | 0.743 | 0.004 | 12.991 | 0.641 |
| Mist (Liang & Wu, 2023) | 0.456 | 0.003 | 18.712 | 0.629 | 0.736 | 0.005 | 25.182 | 0.587 |
| MetaCloak (Liu et al., 2024b) | 0.469 | 0 | 21.861 | 0.599 | 0.727 | 0.007 | 23.027 | 0.618 |
| SimAC (Wang et al., 2024) | 0.495 | 0 | 9.251 | 0.606 | 0.735 | 0.004 | 3.601 | 0.645 |
| DisDiff (Liu et al., 2024a) | 0.535 | 0.003 | 10.252 | 0.601 | 0.708 | 0.021 | 13.488 | 0.639 |
| SDS- (Xue et al., 2023) | 0.499 | 0 | 46.701 | 0.617 | 0.741 | 0.003 | 24.258 | 0.609 |
| PID (Li et al., 2024) | 0.484 | 0.013 | 15.434 | 0.605 | 0.729 | 0.001 | 15.326 | 0.596 |
| LE(ours) | **0.217** | **0.241** | **65.903** | **0.803** | **0.257** | **0.204** | **58.423** | **0.828** |
| LA(ours) | **0.235** | **0.214** | **71.873** | **0.793** | **0.282** | **0.269** | **52.554** | **0.845** |

fusion model architectures. On SD2.1 and SDXL, our methods consistently outperform baselines, achieving the lowest ISM scores (e.g., 0 on SD2.1 and 0.171 on SDXL) and the highest FDFR values (e.g., 1 on SD2.1 and 0.649 on SDXL). This indicates their effectiveness in completely erasing identity semantics while maintaining strong robustness. On models with more advanced architectures like SD3.5 and FLUX.1-dev, which utilize MM-DiT as the denoising network, our methods consistently demonstrate strong performance, achieving significantly lower ISM scores (e.g., 0.211 on SD3.5) and higher Brisque values (e.g., 58.423 on FLUX.1-dev) compared to baselines. These results highlight the adaptability of our methods in handling both traditional UNet-based and advanced denoising architectures, ensuring consistent semantic erasure under various model structures.

**Time and GPU memory consumption:** Considering the limited GPU resources of individual users, we compare the time and GPU memory required by different methods to protect a single image. Table 4 shows that our method uses

only 4.5 GB of GPU memory and completes protection in 8 seconds. In contrast, other methods demand significantly more resources and time. For example, ASPL and SimAC consume over 24 GB of GPU memory, while MetaCloak takes 1843 seconds per image. These results demonstrate the efficiency and practicality of our approach, making it accessible to users with standard setups.

### 4.3. Ablation Experiments

In this section, we present the major ablation experiments. **Additional ablation results and visualizations are provided in the appendix C and D.**

**Attack Step:** Table 5 shows that our methods generate pure noise images with completely erased identity semantics within 30 steps. After 50 steps, the protection behavior changes. The images transition from pure noise to chaotic scene-like visuals, still retaining their ability to erase identity semantics but showing a decline in metric performance.

*Table 4.* Comparison of the time and GPU memory consumption incurred by our method and other approaches to protect a $512 \times 512$ image on A100 GPU.

| Method | Time/s ↓ | GPU/MB ↓ |
|---|---|---|
| AdvDM (Liang et al., 2023) | 18.63 | 8278.63 |
| ASPL (Van Le et al., 2023) | 189.95 | 34366.92 |
| Mist (Liang & Wu, 2023) | 18.81 | 8278.63 |
| MetaCloak (Liu et al., 2024b) | 1843.47 | 16955.00 |
| SimAC (Wang et al., 2024) | 124.57 | 38640.00 |
| DisDiff (Liu et al., 2024a) | 65.54 | 25960.50 |
| SDS- (Xue et al., 2023) | 18.61 | 8278.63 |
| PID (Li et al., 2024) | 241.31 | 4581.93 |
| LE(ours) | **7.34** | **4469.80** |
| LA(ours) | **8.47** | **4469.80** |

*Table 5.* Comparison of defense effectiveness with different iteration steps on CelebA-HQ. "*" is default.

| Step | IMS ↓ | FDFR ↑ | Brisque ↑ | LPIPS ↑ |
|---|---|---|---|---|
| 20 | 0.549 | 0.322 | 56.186 | 0.808 |
| 25 | 0.068 | 0.965 | 123.411 | 1.001 |
| 30* | **0** | **1** | **155.845** | **1.021** |
| 50 | **0** | **1** | 155.692 | 1.019 |
| 75 | 0.022 | 0.993 | 58.244 | 0.933 |
| 100 | 0.019 | 0.997 | 33.288 | 0.912 |

*Table 6.* Ablation study on LE loss terms in CelebA-HQ.

| $\mathcal{L}_{entropy}$ | $\mathcal{L}_{lagrange}$ | IMS ↓ | FDFR ↑ | Brisque ↑ | LPIPS ↑ |
|---|---|---|---|---|---|
| × | × | 0.61 | 0.04 | 17.89 | 0.66 |
| × | ✓ | 0.68 | 0.03 | 17.62 | 0.72 |
| ✓ | × | 0.69 | 0.02 | 23.82 | 0.71 |
| ✓ | ✓ | 0 | 1 | 155.81 | 1.02 |

Therefore, we terminate the optimization at 30 steps. The appendix C.1 provides detailed analysis and visualizations of this behavior.

**Module of LE:** We conduct an ablation experiment on the proposed LE method by testing its performance when the Lagrangian term or the entropy term is removed. As shown in Table 6, using either loss term alone fails to achieve the ideal attack effectiveness. This result demonstrates that the LE loss function is not a simple combination of multiple loss terms but is carefully designed to ensure that the gradient direction aligns with the variance growth direction.

**Noise Budget $\eta$:** We evaluate the impact of the noise budget $\eta$ on the performance of LE on the CelebA-HQ dataset. Table 7 shows that $\eta$=8/255 is sufficient to fully erase identity semantics and produce low-quality images. Increasing $\eta$ to 0.05 raises the Brisque score, suggesting further deviation from natural image characteristics, but without significant visual degradation of adversarial examples. Thus, we set

*Table 7.* Comparison of defense effectiveness and visual quality with different noise budgets $\eta$ on CelebA-HQ. "*" is default.

| $\eta$ | Defense Quality | | Visual Quality | |
|---|---|---|---|---|
| | IMS ↓ | Brisque ↑ | PSNR ↑ | SSIM ↑ |
| 4/255 | 0.631 | 22.385 | 14.244 | 0.414 |
| 8/255 | 0 | 122.266 | 13.771 | 0.361 |
| 0.05* | 0 | 155.804 | 13.664 | 0.313 |
| 16/255 | 0 | 155.845 | 12.301 | 0.271 |

*Table 8.* Comparing the robustness of our methods against image preprocessing and adaptive attack on CelebA-HQ.

| Method | IMS ↓ | FDFR ↑ | Brisque ↑ | LPIPS ↑ |
|---|---|---|---|---|
| Clean | 0.608 | 0.041 | 17.896 | 0.662 |
| Crop | 0.356 | 0.743 | 79.369 | 0.879 |
| Gaussin Filter | 0.369 | 0.697 | 79.607 | 0.913 |
| JPEG Comp. | 0.451 | 0.358 | 62.541 | 0.821 |
| DiffPure | 0.435 | 0.244 | 46.880 | 0.804 |
| GrIDPure | 0.456 | 0.231 | 49.071 | 0.769 |
| Fixed $\sigma$ | 0.503 | 0.255 | 37.920 | 0.823 |
| Clipped $\sigma$ | 0.512 | 0.205 | 39.468 | 0.795 |

$\eta = 0.05$ as the default noise budget.

**Image Preprocessing and Adaptive Attack:** We evaluate the robustness of our methods against image preprocessing and adaptive attacks. For preprocessing, we test traditional methods (Cropping, Gaussian filtering, and JPEG compression), and advanced purification techniques (DiffPure (Nie et al., 2022) and GrIDPure (Zhao et al., 2024)). For adaptive attacks, we simulate scenarios where attackers fix or clip latent variance during fine-tuning to bypass protection. Table 8 shows that our methods still demonstrate notable robustness in these cases. They reduce face detection in generated images (higher FDFR) and lower identity-related similarity (lower IMS). Additionally, the higher Brisque and LPIPS scores indicate a decline in image quality. Although purification and bypassing techniques affect performance, they cannot fully remove our perturbations.

## 5. Conclusion

In this paper, we discuss the relationship between the variance of VAE encoders and the effectiveness of adversarial attacks in disrupting generative diffusion models. We identify variance as key to image distortion and propose two novel loss functions, LA and LE, to overcome limitations of existing methods. These loss functions completely erase identity semantics, generating pure-noise images resistant to misuse. Our methods also excel in transferability and efficiency, requiring fewer resources and achieving faster protection, making them practical for rapidly evolving models and resource-constrained devices.

# Acknowledgments

The authors would like to thank the support of Guangdong Provincial Key Laboratory of Information Security Technology (Grant No.2023B1212060026), Guangdong Provincial Key Laboratory of Intelligent Information Processing and Shenzhen Key Laboratory of Media Security (Grant No.2023B1212060076 ).

# Impact Statement

Our work aims to prevent unauthorized images from being exploited by malicious users to generate fake images through personalized generative techniques. While existing approaches attempt to address this issue, they suffer from several limitations, including slow protection speed, high computational resource requirements, and poor transferability. These drawbacks make them impractical for real-world applications where models undergo rapid iterations or operate in resource-constrained environments. Moreover, most existing methods fail to provide robust identity protection, as they still produce images that retain identity semantics, leaving them vulnerable to misuse. In contrast, our methods effectively remove identity semantics by generating pure noise images, ensuring strong protection against unauthorized identity synthesis while maintaining computational efficiency and transferability across different generative models. Beyond identity protection, our approaches are also applicable to copyright protection in the artistic domain, an increasingly critical issue given the rapid advancement of personalized generative techniques. As AI models become more adept at learning and replicating artistic styles, the risk of unauthorized reproduction and commercialization of artworks grows. Our methods provide proactive defense against such risks by preventing AI models from adapting to protected artistic content. By offering a robust defense mechanism against the unauthorized use of both identity and artistic content, our work contributes to the development of responsible AI, ensuring ethical and secure deployment of generative models.

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

# A. Deep Analysis of Our Work

## A.1. Rationale for Neglecting Higher-Order Terms under Small Perturbations

In adversarial scenarios, we often examine how a small perturbation $\delta \in \mathbb{R}^n$ propagates through a deep network (e.g., the VAE encoder in Stable Diffusion) and affects the final loss. A common strategy is to linearize the network around the clean input $x$, effectively discarding higher-order (nonlinear) terms. In the following, we provide a step-by-step justification for ignoring these higher-order corrections when $\|\delta\| \leq \epsilon$ is sufficiently small.

**Second-Order Taylor Expansion.** Let $f : \mathbb{R}^n \to \mathbb{R}^m$ denote the VAE encoder, mapping an input $x$ to a latent representation $z = f(x)$. A second-order Taylor expansion around $x$ yields:

$$f(x + \delta) = f(x) + J_f(x)\,\delta + \tfrac{1}{2}\,\delta^\top H_f\big(x + \theta\delta\big)\,\delta, \quad 0 \leq \theta \leq 1, \tag{19}$$

where $J_f(x) = \nabla_x f(x)$ is the Jacobian matrix, and $H_f(\cdot)$ is the Hessian. The remainder term $\tfrac{1}{2}\,\delta^\top H_f(\cdot)\,\delta$ represents second-order (and beyond) corrections.

**$\beta$-Smoothness Assumption.** A standard assumption in optimization is that $f$ is $\beta$-*smooth* (or has Lipschitz continuous gradients) within a local neighborhood of $x$ (Nesterov, 2013). Concretely,

$$\|\nabla f(u) - \nabla f(v)\| \leq \beta \|u - v\|, \quad \forall\, u, v \in \mathcal{N}(x). \tag{20}$$

Equivalently, the Hessian operator norm satisfies $\|H_f(z)\| \leq \beta$. Under this assumption, we can bound the second-order remainder in (19) by

$$\left\|\tfrac{1}{2}\,\delta^\top H_f(x + \theta\delta)\,\delta\right\| \leq \tfrac{\beta}{2}\,\|\delta\|^2. \tag{21}$$

Hence, if $\|\delta\| \leq \epsilon$ is small, the contribution from higher-order terms scales as $O(\epsilon^2)$, which is typically negligible compared to the first-order term $O(\epsilon)$.

**Influence on the Loss Gradient.** In adversarial settings, the primary concern is often $\nabla_\delta \mathcal{L}\big(f(x + \delta)\big)$. By the chain rule,

$$\nabla_\delta \mathcal{L}\big(f(x + \delta)\big) = \big[\nabla_z \mathcal{L}(z)\big]_{z = f(x + \delta)} \cdot \nabla_\delta f(x + \delta). \tag{22}$$

With the $\beta$-smooth assumption, one can show the discrepancy between the exact gradient and its first-order approximation is bounded by $O(\|\delta\|)$ (Goodfellow et al., 2015). Therefore, in a sufficiently small neighborhood, ignoring higher-order terms does not significantly alter the gradient direction that guides adversarial perturbations.

**Consistency with Iterative Attacks.** Projected Gradient Descent (PGD) (Madry et al., 2018) updates $\delta$ in small increments,

$$\delta \leftarrow \Pi_{\|\delta\| \leq \epsilon}\Big(\delta + \alpha\,\nabla_\delta \mathcal{L}\big(f(x + \delta)\big)\Big), \tag{23}$$

ensuring that the input $x + \delta$ remains within the local region $\|\delta\| \leq \epsilon$. Such small-step iterative methods naturally align with a local linear approximation, as they do not abruptly jump outside the domain where $\beta$-smoothness is valid. Hence, the ignored second-order terms remain bounded throughout the attack process.

**Empirical Observations.** Many adversarial methods, including PGD (Madry et al., 2018) and C&W (Carlini & Wagner, 2017), rely almost exclusively on first-order information yet demonstrate highly effective attacks in small-$\|\delta\|$ regimes. Empirical comparisons further suggest that the angle between the exact gradient and the first-order approximation remains small (Goodfellow et al., 2015), reinforcing the idea that higher-order terms can be safely neglected for small perturbations.

**Conclusion.** Under the $\beta$-*smoothness* hypothesis and for perturbations constrained by $\|\delta\| \leq \epsilon$, the second-order remainder in (19) contributes only $O(\epsilon^2)$, hence can be dismissed relative to the $O(\epsilon)$ first-order effect. Consequently, linearizing the VAE encoder (or other neural components) is both a common heuristic and a theoretically supported practice in adversarial analysis.

## A.2. Jacobian Computation: Practical Challenges and Mitigation

In our theoretical derivations (see Eqs.8–10), we decompose how a perturbation $\delta$ propagates through the VAE encoder using Jacobian matrices. However, explicitly computing these Jacobians in practice can face several challenges:

**High Dimensionality and Computation Costs.** For inputs of size $H \times W \times 3$ (e.g., color images) and a high-dimensional latent space in the VAE, the Jacobian matrix can be extremely large. Storing or multiplying such matrices directly would require substantial memory and computation. When either the input resolution or the latent-space dimension grows, the overall Jacobian can easily scale to millions of parameters.

**No Need for Exact Elementwise Computation.** In our formulation, the purpose of these Jacobians is primarily theoretical to show how a perturbation flows through successive layers and eventually affects the variance term $\sigma^2$ and the final loss. We **do not** advocate explicit elementwise computation or storage of such high-dimensional matrices in actual implementations. Instead, we mainly rely on the chain structure of the Jacobian product to reveal how each layer's local derivatives combine. In practice, automatic differentiation typically yields the end-to-end gradient without needing to expand the intermediate Jacobians. Moreover, as demonstrated in Table 4, our method only requires approximately 4.5 GB of GPU memory on an NVIDIA A100 to protect a single $512{\times}512$ image in about 8 seconds, underscoring that we do not need to hold the entire Jacobian in memory to achieve efficient and feasible runtime.

**Gradient Stability (See Section C.1).** Deep networks are known to exhibit exploding or vanishing gradients, especially when certain activation functions saturate or become inactive. Although this phenomenon is not strictly caused by explicitly handling the Jacobian, it can affect the numerical stability of Jacobian-based products. Moreover, gradient masking can sometimes occur when the network's gradient pathways are obstructed or improperly managed (e.g., due to overly aggressive regularization or non-differentiable structures). To mitigate these issues, we employ an early-stopping strategy during the optimization steps (see Section C.1). This early-stopping procedure helps avoid practical pitfalls such as exploding or vanishing gradients and reduces the risk of artificially masking the true gradient signals.

**Summary.** In short, the Jacobians in Eqs. 8-10 primarily serve an analytical role: they illustrate how perturbations impact each stage of the encoder in a chain-of-influence manner. These do not imply that one must explicitly compute or store the entire Jacobian in a real-world scenario. **Our experimental results indicate that our method requires only about 4.5 GB of GPU memory on an NVIDIA A100 to protect a single $512{\times}512$ image in roughly 8 seconds,** underscoring that it is unnecessary to hold the full Jacobian in memory to achieve efficient and feasible runtime performance.

## A.3. Illustrative Example for Gradient Sign Reversal

Before we delve into how the Laplace Approximation (LA) maintains a consistent variance-growth direction, let us first consider a concrete numerical example to illustrate how the gradient direction (as computed via Eq. 11) can be influenced by the loss derivatives $\frac{\partial \mathcal{L}}{\partial \sigma_j^2}$ in MSE, PID, LA and LE. To illustrate how each loss behaves when certain variances already exceed the clean variance $\sigma_{\text{clean}}^2$, consider a simple three-dimensional scenario:

$$\sigma^2 = \left[\sigma_1^2,\ \sigma_2^2,\ \sigma_3^2\right] = [0.001,\ 0.001,\ 0.006], \quad \sigma_{\text{clean}}^2 \ll \{0.001,\ 0.006\}. \tag{24}$$

Furthermore, let the Jacobian for a single perturbation dimension $\delta_i$ be

$$\frac{\partial \sigma_j^2}{\partial \delta_i} = M_{i,j} = [1,\ 1,\ -1]. \tag{25}$$

Note that $\sigma_3^2$ is notably larger than $\sigma_{\text{clean}}^2$ and also has a negative sensitivity $-1$. The emergence of such an extreme variance component like $\sigma_3^2 = 0.006$ can be attributed to the influence of the initial perturbation. Initially, the variance matrix is uniformly distributed. However, the perturbation disrupts this balance, causing the variance distribution to become uneven.

**(1) MSE Loss.** We define

$$\mathcal{L}_{MSE} = \sum_j \left(\sigma_j^2 - \sigma_{clean}^2\right)^2, \quad \frac{\partial \mathcal{L}_{MSE}}{\partial \sigma_j^2} = 2\left(\sigma_j^2 - \sigma_{clean}^2\right). \tag{26}$$

Substituting $\sigma_1^2 = 0.001$, $\sigma_2^2 = 0.001$, $\sigma_3^2 = 0.006$ and assuming $\sigma_{clean}^2$ is much smaller (effectively zero compared to these values), yields partial derivatives $2 \times 0.001$, $2 \times 0.001$, $2 \times 0.006 = 0.002$, $0.002$, $0.012$. Hence,

$$\frac{\partial \mathcal{L}_{MSE}}{\partial \delta_i} = \sum_{j=1}^{3} \left[ 2(\sigma_j^2 - \sigma_{clean}^2) \right] \times (M_{i,j}) = 0.002 \cdot 1 + 0.002 \cdot 1 + 0.012 \cdot (-1) = -0.008, \tag{27}$$

which is **negative**. Although each $\sigma_j^2$ term is larger than $\sigma_{clean}^2$, the third component's combination of a larger partial derivative $(0.012)$ and negative Jacobian factor $(-1)$ dominates, reversing the overall gradient sign.

Therefore, MSE prioritizes optimizing large variance components (e.g., $0.006$) while neglecting smaller variance components (e.g., $0.001$). As a result, the gradient sign of MSE is often misaligned with the direction of increasing variance $M_{i,j}$. This misalignment causes MSE to fall into a local optimization trap, where it fails to effectively increase either its targeted large variance components or the smaller variance components that require growth.

**(2) PID Loss.** Define

$$\mathcal{L}_{PID} = \sum_{j} \left( \log \sigma_j^2 - \log \sigma_{clean}^2 \right)^2, \quad \frac{\partial \mathcal{L}_{PID}}{\partial \sigma_j^2} = \frac{2}{\sigma_j^2} \left( \log \sigma_j^2 - \log \sigma_{clean}^2 \right). \tag{28}$$

For $\sigma_3^2 = 0.006 \gg \sigma_{clean}^2$, the term $\log(\sigma_3^2) - \log(\sigma_{clean}^2)$ is positive, ensuring that $\frac{\partial \mathcal{L}_{PID}}{\partial \sigma_3^2} > 0$. Thus, PID maintains a positive contribution to $\frac{\partial \mathcal{L}}{\partial \delta_i}$ unless the negative Jacobian factor is extremely large. This means that PID correctly prioritizes small variance components during the initial optimization, where gradients are larger for smaller variances due to the $1/\sigma_j^2$ factor.

However, as small variance components are optimized and their values grow, the variance matrix becomes more uniform. This uniformity reduces the optimization space, as the differences between variance components shrink. Additionally, the PID gradient $\frac{\partial \mathcal{L}_{PID}}{\partial \sigma_j^2}$ weakens over time because the $\frac{1}{\sigma_j^2}$ term diminishes as $\sigma_j^2$ increases. Consequently, PID requires significantly more steps to achieve sufficiently large variance. This slow convergence is reflected in its original work, where PID requires 900 steps to remove most identity semantics, compared to the 30–50 steps required by our proposed methods.

**(3) LA (Laplace Approx.) Loss.** The Laplace Approximation (LA) Loss is defined as:

$$\mathcal{L}_{LA} = \frac{|\sigma^2 - \mu|}{b}, \quad \frac{\partial \mathcal{L}_{LA}}{\partial \sigma_j^2} = \frac{1}{b}, \tag{29}$$

where $\mu$ is typically set to $0$, and $b$ is a scaling factor, typically set to $1$. Importantly, the gradient with respect to $\sigma_j^2$ is constant and independent of the variance values, meaning the optimization direction is determined solely by the mapping relationship $M_{i,j}$.

For the example where $M_{i,j} = [1, 1, -1]$, the contribution to $\frac{\partial \mathcal{L}}{\partial \delta_i}$ becomes:

$$\frac{\partial \mathcal{L}_{LA}}{\partial \delta_i} = \sum_{j} \frac{\partial \mathcal{L}_{LA}}{\partial \sigma_j^2} \cdot M_{i,j} = \frac{1}{b} \cdot \sum_{j} M_{i,j} = \frac{1}{b}(1 + 1 - 1) = \frac{1}{b}. \tag{30}$$

Here, $\frac{\partial \mathcal{L}_{LA}}{\partial \delta_i}$ is guaranteed to remain positive since $b > 0$. Unlike MSE or PID, LA loss maintains a consistent alignment with the variance growth direction, as the sign of the gradient depends only on the structure of $M_{i,j}$, and there are no cancellations caused by varying magnitudes of $\sigma_j^2$ or $\frac{\partial \mathcal{L}}{\partial \sigma_j^2}$.

Moreover, this constant gradient ensures simultaneous and balanced growth of variance across all dimensions. Unlike PID, which prioritizes small variances but slows down as variance grows, or MSE, which prioritizes large variances but may reverse gradient direction, LA Loss provides consistent updates in the direction of increasing variance for all components. This makes LA Loss robust and efficient, avoiding the pitfalls of gradient misalignment or diminishing optimization space.

In addition to its consistent direction, LA Loss maintains a large optimization space throughout the process. The constant gradient $\frac{\partial \mathcal{L}_{LA}}{\partial \sigma_j^2} = \frac{1}{b}$ ensures equal optimization for all variance components, regardless of their initial magnitudes. Unlike

PID, where the gradient diminishes as variance grows, LA Loss sustains fixed optimization strength across both small and large variances, preventing stagnation or bias toward specific components. This balanced optimization accelerates variance growth by enabling simultaneous updates across all components. As a result, LA Loss rapidly amplifies variance and removes identity semantics within 30–50 steps, significantly outperforming PID's 900-step optimization.

**(4) LE Loss**   Recall that our Lagrange Entropy (LE) Loss is given by

$$\mathcal{L}_{LE}(\sigma^2) = -\sum_j \sigma_j^2 \log(\sigma_j^2) + \lambda \left( \sum_j \sigma_j^2 - c \right)^2. \tag{31}$$

When $c = 1$,

$$\mathcal{L}_{LE}(\sigma^2) = -\sum_j \sigma_j^2 \log(\sigma_j^2) + \lambda \left( \sum_j \sigma_j^2 - 1 \right)^2, \tag{32}$$

whose partial derivative reads

$$\frac{\partial \mathcal{L}_{LE}}{\partial \sigma_j^2} = -\big[\log(\sigma_j^2) + 1\big] + 2\lambda \Big( \sum_k \sigma_k^2 - 1 \Big). \tag{33}$$

We can interpret the first part $-\big[\log(\sigma_j^2) + 1\big]$ as an *entropy-driven term* acting on each dimension, while $2\lambda(\sum_k \sigma_k^2 - 1)$ acts as a *global Lagrange penalty* controlling the sum of variances.

**Note on logarithms.** Throughout this paper (and in our code), $\log(\cdot)$ refers to the natural logarithm $\ln(\cdot)$.

**Case 1: When $\sum_j \sigma_j^2 < 1$, the entropy term dominates (positive) while the Lagrange penalty is negative.**   If $\sigma_j^2 < 1$, then $\log(\sigma_j^2) < 0$, so $-\big[\log(\sigma_j^2) + 1\big]$ is a significantly positive quantity. For example, if $\sigma_j^2 = 0.01$, $\log(0.01) \approx -4.605$ and $-[\log(0.01) + 1] \approx +3.605$, giving a strong boost to that dimension. Meanwhile, because $\sum_j \sigma_j^2 - 1 < 0$, the Lagrange term $2\lambda(\sum_j \sigma_j^2 - 1)$ is negative, partially offsetting the entropy push. However, while the total variance is still well below 1, most dimensions have $\sigma_j^2 < 1$, so multiple positive entropy terms outweigh this negative penalty overall. Thus, the net gradient remains positive, continuing to increase $\sigma_j^2$ without flipping sign or stalling.

**Case 2: When $\sum_j \sigma_j^2 > 1$, the Lagrange penalty dominates (positive) while the entropy term can be negative.**   For dimensions whose $\sigma_j^2 > 1$, $\log(\sigma_j^2) + 1$ is positive, so $-\big[\log(\sigma_j^2) + 1\big]$ becomes negative, preventing any single dimension from growing unboundedly. On the other hand, if $\sum_j \sigma_j^2 - 1 > 0$, $2\lambda(\sum_j \sigma_j^2 - 1)$ is a positive addition. A sufficiently large positive penalty can override the negative entropy factor. For example, if $\log(\sigma_j^2) + 1 = 1.693$ (for $\sigma_j^2 \approx 2$), then the negative part is $-1.693$, but if $2\lambda(\sum_j \sigma_j^2 - 1) = +3$, the sum is still $+1.307$, ensuring a net positive gradient. As a result, even large variances do not cause the update direction to flip to negative. The global Lagrange term ensures a consistent push if the total variance is meant to exceed 1.

**Implications for Optimization Space and Speed.**

- **No severe compression**: A common pitfall in PID is that $\frac{\partial \mathcal{L}_{PID}}{\partial \sigma_j^2} \propto \frac{1}{\sigma_j^2}$ diminishes severely for large $\sigma_j^2$, slowing down further variance increases. In contrast, LE's entropy part $-\sigma^2 \log(\sigma^2)$ varies only logarithmically with $\sigma^2$. As $\sigma_j^2$ grows from 0.1 to 1 or even 5, $\log(\sigma_j^2)$ changes moderately, so $-\big[\log(\sigma_j^2) + 1\big]$ remains in a range that does not collapse to near-zero magnitude. Furthermore, the Lagrange term $\lambda(\sum_j \sigma_j^2 - 1)^2$ adds a finite amount, but does not drastically shrink the net gradient. Thus, LE avoids the late-stage gradient exhaustion seen in PID and also avoids MSE's extreme sign flips caused by large outlier variances.

- **Minimal risk of sign inversion**: MSE can create large partial derivatives that, if multiplied by negative $\frac{\partial \sigma_j^2}{\partial \delta}$, flip the total sign. In LE Loss, the interplay of a negative entropy term (for big $\sigma_j^2$) and a positive Lagrange term often cancels out in favor of a mild positive net gradient.

- **LA-like speed**: LA uses a constant gradient independent of $\sigma_j^2$, quickly boosting variances in 30–50 steps. LE, with the two-phase dominance (entropy vs. Lagrange), similarly avoids any late-stage slowdown or early-stage stalling. Thus, it can achieve variance expansion nearly as swiftly as LA.

**Conclusion.** In this three-dimensional example, MSE can yield a negative overall gradient (despite $\sigma_j^2 > \sigma_{clean}^2$) due to the mismatch with the negative Jacobian in the third dimension. PID does avoid sign flips but loses momentum for larger variances, thus requiring many iterations to further increase $\sigma^2$. LA, by contrast, keeps a constant gradient that remains aligned with the "increase-variance" direction at every step. Meanwhile, LE combines the advantages of LA's stable gradient with an additional entropy and Lagrange-based balancing. It achieves near-constant or steadily positive updates while regulating extreme growth in any single variance dimension. Hence, LE can match LA's rapid speed of variance expansion and avoid both MSE's sign flips and PID's late-stage slowdown.

### A.4. Why Only Attack Variance Rather Than Mean?

PID (Li et al., 2024)) observes that attacking the mean $\mu$ of the VAE encoder primarily introduces noisy artifacts or textures into the generated images. Although these artifacts become more pronounced with additional attack steps, they do not effectively remove identity or semantic information. In contrast, attacking the variance $\sigma^2$ directly impacts the diffusion model's ability to extract semantic concepts. As documented by PID, increasing the variance optimization steps (from 0 to 300, 600, or even 900 steps) gradually erase the identity semantics in the generated image until it is fully obfuscated.

Hence, to remove high-level semantics (e.g., identity), it is sufficient to focus on $\frac{\partial \mathcal{L}}{\partial \sigma^2}$ and ignore the mean. The mean perturbation adds little to semantic destruction besides additional noise-like artifacts. Therefore, in our chain-rule analysis, we primarily consider how $\sigma^2$ evolves under adversarial perturbations $\delta$, rather than factoring in the mean term.

*Table 9.* Comparing the performance of our method with baselines against DreamBooth (Ruiz et al., 2023) on CelebA-HQ and VGGFace2. The best result under each metric is marked with **bold**. The prompt used is *"a dslr portrait of sks person."*

| Method | CelebA-HQ | | | | VGGFace2 | | | |
|---|---|---|---|---|---|---|---|---|
| | ISM ↓ | FDFR ↑ | Brisque ↑ | LPIPS ↑ | ISM ↓ | FDFR ↑ | Brisque ↑ | LPIPS ↑ |
| No Defense | 0.419 | 0.085 | 7.506 | 0.748 | 0.459 | 0.048 | 9.496 | 0.769 |
| AdvDM (Liang et al., 2023) | 0.319 | 0.067 | 11.461 | 0.796 | 0.263 | 0.139 | 15.288 | 0.821 |
| ASPL (Van Le et al., 2023) | 0.316 | 0.064 | 11.381 | 0.802 | 0.289 | 0.113 | 14.195 | 0.819 |
| Mist (Liang & Wu, 2023) | 0.071 | 0.095 | 13.338 | 0.864 | 0.153 | 0.078 | 17.497 | 0.857 |
| MetaCloak (Liu et al., 2024b) | 0.358 | 0.117 | 25.51 | 0.781 | 0.377 | 0.062 | 20.102 | 0.803 |
| SimAC (Wang et al., 2024) | 0.328 | 0.298 | 18.143 | 0.776 | 0.384 | 0.263 | 19.618 | 0.787 |
| DisDiff (Liu et al., 2024a) | 0.396 | 0.049 | 48.672 | 0.749 | 0.376 | 0.254 | 17.863 | 0.821 |
| SDS- (Liu et al., 2024a) | 0.378 | 0.028 | 9.435 | 0.805 | 0.414 | 0.033 | 37.508 | 0.825 |
| PID (Li et al., 2024) | 0.077 | 0.631 | 35.317 | 0.899 | 0.077 | 0.791 | 48.318 | 0.928 |
| LE(ours) | **0.031** | **0.918** | **103.004** | **0.968** | **0.044** | **0.849** | **91.221** | **0.967** |
| LA(ours) | **0.034** | **0.827** | **96.181** | **0.959** | **0.056** | **0.867** | **86.236** | **0.974** |

*Table 10.* Comparison of our method with other approaches against Textual Inversion and IP-Adapter on CelebA-HQ. The best-performing defense under each metric is marked with **bold**.

| Method | Textual Inversion | | | | IP-Adapter | | | |
|---|---|---|---|---|---|---|---|---|
| | ISM ↓ | FDFR ↑ | Brisque ↑ | LPIPS ↑ | ISM ↓ | FDFR ↑ | Brisque ↑ | LPIPS ↑ |
| No Defense | 0.567 | 0.014 | 21.432 | 0.512 | 0.341 | 0.025 | 18.436 | 0.689 |
| AdvDM (Liang et al., 2023) | 0.292 | 0.358 | 43.309 | 0.818 | 0.319 | 0.021 | 40.057 | 0.768 |
| ASPL (Van Le et al., 2023) | 0.332 | 0.268 | 40.235 | 0.812 | 0.328 | 0.017 | 40.941 | 0.768 |
| Mist (Liang & Wu, 2023) | 0.106 | 0.795 | 28.514 | 0.743 | 0.311 | 0.026 | 32.229 | 0.834 |
| MetaCloak (Liu et al., 2024b) | 0.231 | 0.221 | 42.588 | 0.755 | 0.305 | 0.028 | 37.003 | 0.785 |
| SimAC (Wang et al., 2024) | 0.122 | 0.792 | 67.545 | 0.843 | 0.317 | 0.028 | 41.338 | 0.753 |
| DisDiff (Liu et al., 2024a) | 0.345 | 0.251 | 35.038 | 0.770 | 0.319 | 0.018 | 21.866 | 0.747 |
| SDS- (Xue et al., 2023) | 0.452 | 0.057 | 44.240 | 0.764 | 0.296 | 0.022 | 29.968 | 0.806 |
| PID (Li et al., 2024) | 0.054 | 0.938 | 82.314 | 0.926 | 0.296 | 0.018 | 21.568 | 0.819 |
| LE(ours) | **0.017** | **0.988** | **89.537** | **0.958** | **0.078** | **0.248** | **47.937** | **0.875** |
| LA(ours) | **0.014** | **0.958** | **87.349** | **0.943** | **0.091** | **0.232** | **45.796** | **0.893** |

# B. More Quantitative Results

## B.1. Quantitative Results for mismatch prompt against DreamBooth

We simulate a scenario where a user, after training a DreamBooth model, employs an inference prompt *"a dslr portrait of sks person,"* which differs from the training prompt *"a photo of a sks person."* The specific results are presented in Table 9 and Figure 6. We observe that when the inference prompt does not match the training prompt, the identity similarity metric (ISM) of the baselines decreases. Meanwhile, the noise in the generated images is significantly reduced, as reflected in the Brisque and LPIPS scores of the baselines, which become similar to those of the No Defense method. In contrast, our approaches consistently produce pure noise images, demonstrating greater robustness.

## B.2. Quantitative Results for Textual Inversion and IP-Adapter

We report the results of using Textual Inversion (Gal et al., 2022) and IP-Adapter (Ye et al., 2023) as personalization algorithms on the CelebA-HQ dataset in Table 10. Our methods outperform other methods in resisting Textual Inversion and IP-Adapter. For Textual Inversion, we consistently maintain extremely low ISM scores and high FDFR scores, demonstrating that the generated images contain almost no recognizable facial semantics. When applied to IP-Adapter, only our methods achieve an ISM score below 0.1 and an FDFR score above 0.2. These results confirm that our methods are well-suited to resist various mainstream personalization generation algorithms.

# C. More Ablation Experiments

## C.1. Attack step

**Gradient Explosion and Unstable Oscillations.** We conduct an ablation study on the number of optimization steps for LE and LA and observe a notable phenomenon in Figure 4(a): $\frac{\partial \mathcal{L}_{LE}}{\partial \delta}$ experiences a gradient explosion after 30 steps, whereas $\frac{\partial \mathcal{L}_{LA}}{\partial \delta}$ begins to oscillate unstably around 50 steps. We hypothesize that these instabilities arise from the product of the two Jacobians ($J_f$ mapping $\delta$ to hidden feature $\phi$ and $J_g$ mapping $\phi$ to $\sigma^2$), amplified by the nonlinearities in the network. Once certain activation functions saturate or shift during high-dimensional updates, the gradients can grow or fluctuate dramatically as the number of steps increases. Figure 4(b) illustrates the generated images at various optimization steps. Between 30 and 75 steps, both LE and LA produce nearly pure noise. When the step count exceeds 100, the latent distribution drifts further, resulting in chaotic "scene-like" patterns. As reported in Table 5, while such images exhibit degraded metrics they nonetheless fully remove identity semantics, thus providing effective protection.

## C.2. Uncontrolled Settings:

Our methods introduce carefully crafted perturbations to images to prevent diffusion models from learning their features, and these perturbations are applied to all images in our experiments. In this section, we assume that malicious users can access some clean images and mix them with the protected ones as the training set. Table 11 demonstrates that even under different protection ratios, our methods remain effective.

Table 11. Comparison between perturbed images and clean images at different ratios on CelebA-HQ.

| Perturbed | Clean | IMS ↓ | FDFR ↑ | Brisque ↑ | LPIPS ↑ |
|---|---|---|---|---|---|
| 0 | 4 | 0.608 | 0.041 | 17.896 | 0.662 |
| 1 | 3 | 0.536 | 0.148 | 26.449 | 0.771 |
| 2 | 2 | 0.481 | 0.191 | 44.075 | 0.821 |
| 3 | 1 | 0.227 | 0.916 | 123.387 | 0.972 |
| 4 | 0 | 0 | 1 | 155.804 | 1.021 |

## C.3. On WikiArt

We further evaluate the effectiveness of our methods in protecting artistic works. We select five different styles from the WikiArt dataset (Saleh & Elgammal, 2015): Baroque, Cubism, Expressionism, Fauvism, and Romanticism. For each style, we use four paintings as the training set for DreamBooth, with the training prompt set to *"a photo of sks picture"*. As shown in Figure 13, both our LA and LE methods successfully generate pure noise images, effectively preventing the imitation of

$$\frac{\partial L_{LE}}{\partial \delta}$$

$$\frac{\partial L_{LA}}{\partial \delta}$$

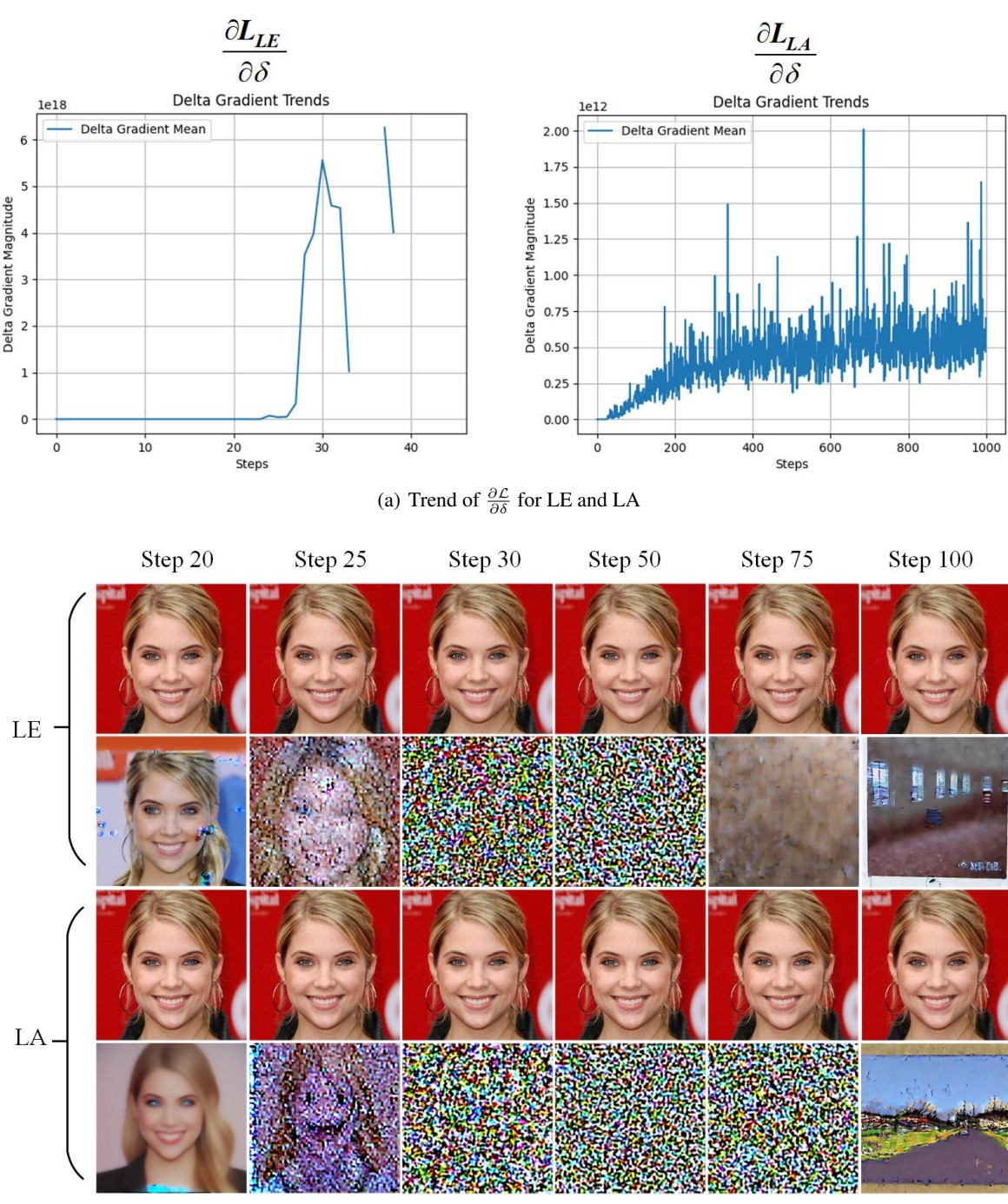

(a) Trend of $\frac{\partial \mathcal{L}}{\partial \delta}$ for LE and LA

(b) Visualization of generated images under different optimization steps for LE and LA Loss.

Figure 4. (a) Trend of $\frac{\partial \mathcal{L}}{\partial \delta}$ for LE and LA as optimization steps increase. The gradient of LE explodes after step 30, while the gradient of LA becomes unstable after step 50. (b) Adversarial effects of LE and LA at different steps. Since both LE and LA generate pure noise images by step 30, we apply an early-stopping strategy to prevent gradient issues and reduce unnecessary computational overhead.

artistic styles. This demonstrates that our approach can be extended beyond identity protection and applied to safeguarding artistic creations from unauthorized style replication.

## D. More Visualization

### D.1. Visualization of Transferability

We present visualizations in Figures 7-10, demonstrating the transferability of adversarial examples generated by our methods and baselines on Stable Diffusion v1.5 to other models. Specifically, we fine-tune LoRA using these adversarial examples as the training set on Stable Diffusion 2.1 (SD2.1), SDXL (Podell et al., 2024), Stable Diffusion 3.5 Medium (SD3.5) (Esser et al., 2024), and FLUX.1-dev (Labs, 2023), and analyze the resulting images. As shown in Figure 7, since SD2.1 shares a nearly identical architecture with SD1.5, both our methods and baselines maintain strong protective effects. The baselines introduce distortions and partially remove semantic content, while our method completely eliminates identity semantics, generating pure noise images. However, the protection effectiveness decreases with SDXL, SD3.5, and FLUX.1-dev due to significant architectural differences from SD1.5.

**Architectural Differences:**

**SDXL vs. SD1.5:** Unlike SD1.5, which uses a single UNet, SDXL adopts a two-stage UNet architecture with a base UNet followed by a refiner UNet for improved image quality. Additionally, SDXL employs a larger text encoder (CLIP ViT-L/14) and improved conditioning mechanisms, making it more resistant to perturbations designed for SD1.5.

**SD3.5 and FLUX.1-dev:** These models replace the traditional UNet with a more advanced Multi-Modal Diffusion Transformer (MM-DiT) architecture, which significantly alters the way latent representations are processed. MM-DiT improves efficiency, handles multimodal inputs, and leverages self-attention across broader spatial contexts, reducing the impact of attacks targeting UNet structures.

Due to these changes, methods that rely on UNet gradient information (e.g., AdvDM (Liang et al., 2023), ASPL (Van Le et al., 2023), MetaCloak (Liu et al., 2024b), SimAC (Wang et al., 2024), SDS- (Xue et al., 2023)) or cross-attention manipulation (e.g., DisDiff (Liu et al., 2024a)) largely fail to maintain protection, with even noticeable noise artifacts disappearing. On the other hand, Mist (Liang & Wu, 2023) and PID (Li et al., 2024), which attack the VAE module, retain some protective effects by introducing visible noise distortions. However, the generated images still preserve identity semantics, limiting their effectiveness.

In contrast, our methods continue to eliminate most identity semantics, demonstrating superior transferability and practical applicability across different model architectures.

### D.2. Against ControlNet-based Image Edit

We evaluate our methods and baselines in resisting ControlNet-based (Zhang et al., 2023) image editing. Specifically, we used ControlNet v1.1 with Depth Map, SoftEdge, OpenPose, Normal Map, and Segmentation as conditional guidance for image editing. The visualization results are presented in Figure 12. As observed, baselines continue to add varying levels of noise to the generated images, whereas our method completely removes identity semantics, producing pure noise images that cannot be used for unauthorized purposes.

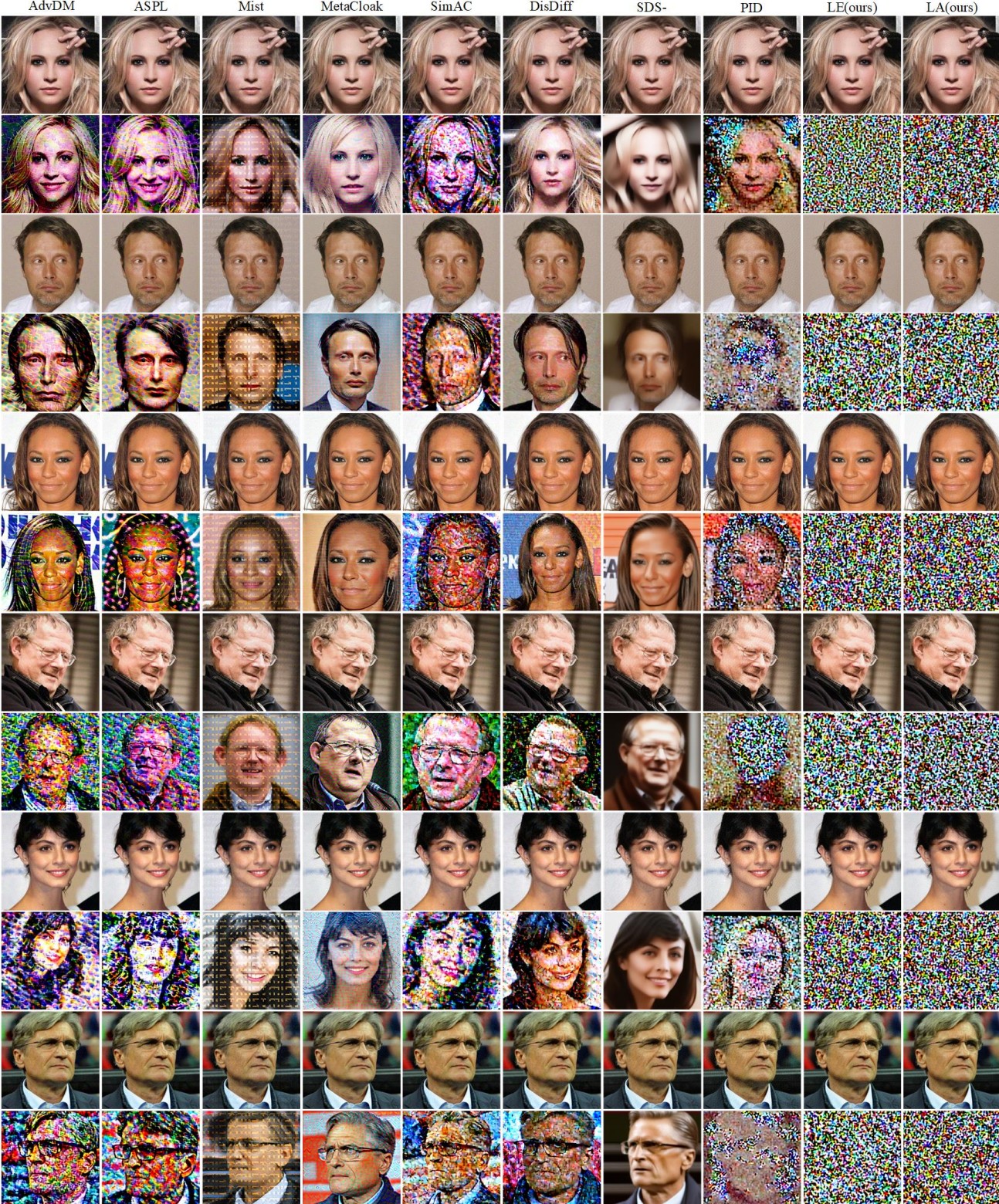

*Figure 5.* More visualization of our methods and baselines against **DreamBooth** on CelebA-HQ and VGGFace2. The odd-numbered rows show reference images, while the even-numbered rows display generated images. The inference prompt used is *"a photo of a sks person."* All baselines add different levels of noise to the generated images, while only our methods completely remove identity semantics and generate pure noise images.

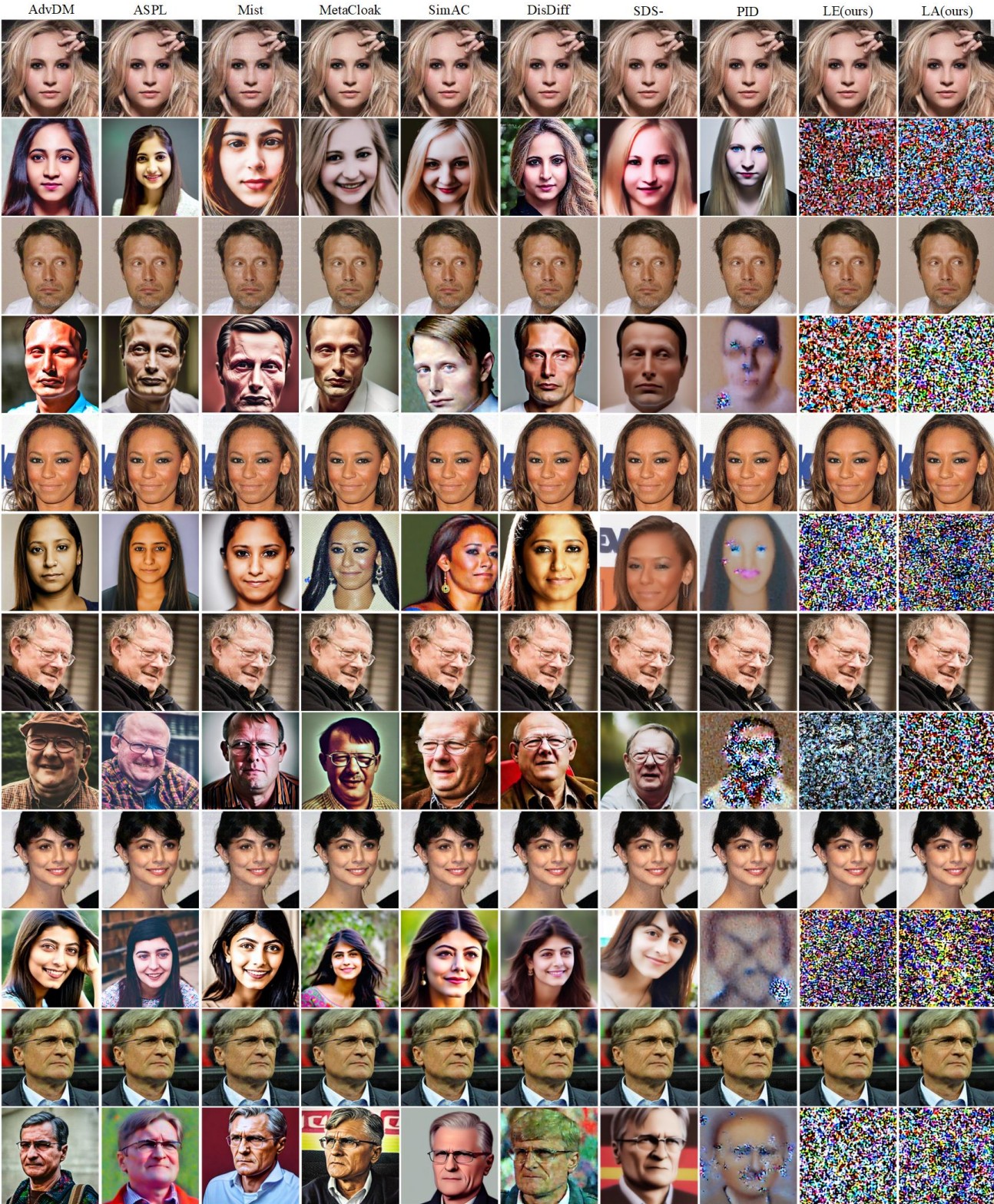

*Figure 6.* More visualization of our methods and baselines against **DreamBooth** on CelebA-HQ and VGGFace2. The odd-numbered rows show reference images, while the even-numbered rows display generated images. The inference prompt used is *"a dlsr portrait of sks person."* The performance of baselines is affected by the mismatch between the inference prompt and the training prompt, while our methods still completely remove identity semantics and generate pure noise images. Best viewd in zoom.

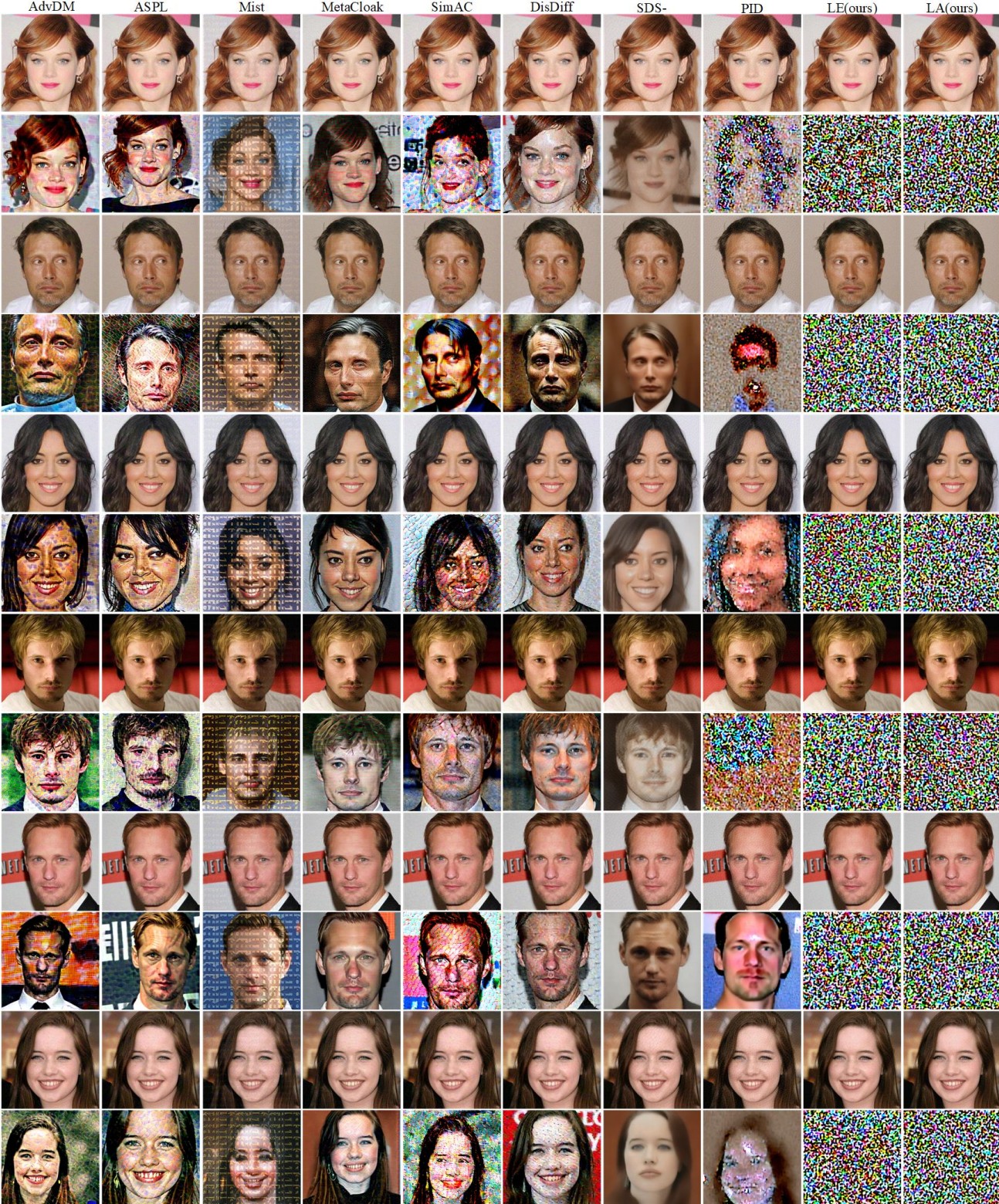

*Figure 7.* More visualization of our methods and baselines against **LoRA** on CelebA-HQ. The odd-numbered rows show reference images, while the even-numbered rows display generated images. Adversarial examples generated on SD1.5 are transferred to **SD2** as the training set. The noise originally added by the baselines is largely removed, indicating poor transferability. In contrast, our method still generates images that are unrelated to identity semantics, demonstrating better transferability. Best viewd in zoom.

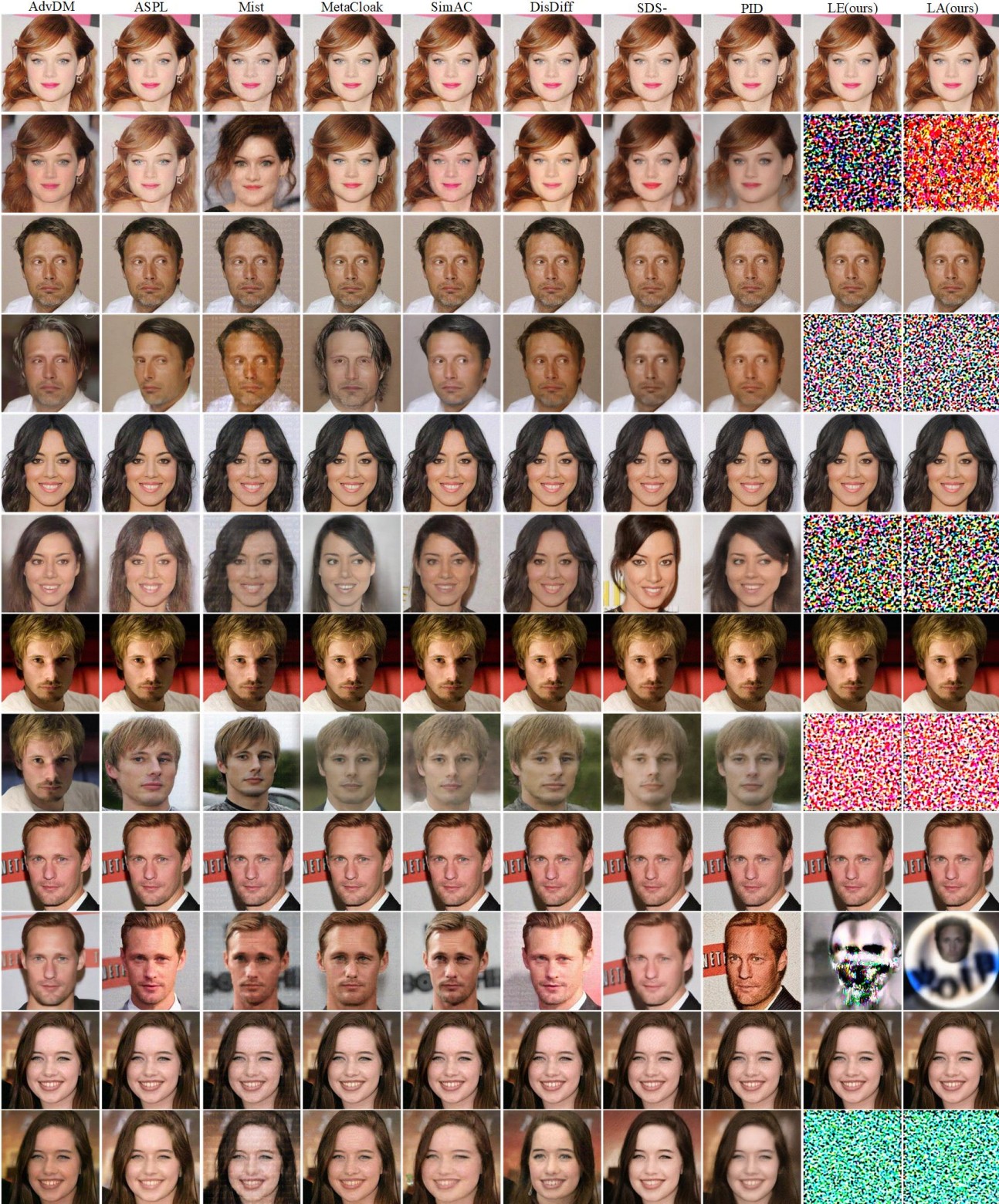

*Figure 8.* More visualization of our methods and baselines against **LoRA** on CelebA-HQ. The odd-numbered rows show reference images, while the even-numbered rows display generated images. Adversarial examples generated on SD1.5 are transferred to **SDXL** as the training set. The noise originally added by the baselines is largely removed, indicating poor transferability. In contrast, our method still generates images that are unrelated to identity semantics, demonstrating better transferability. Best viewd in zoom.

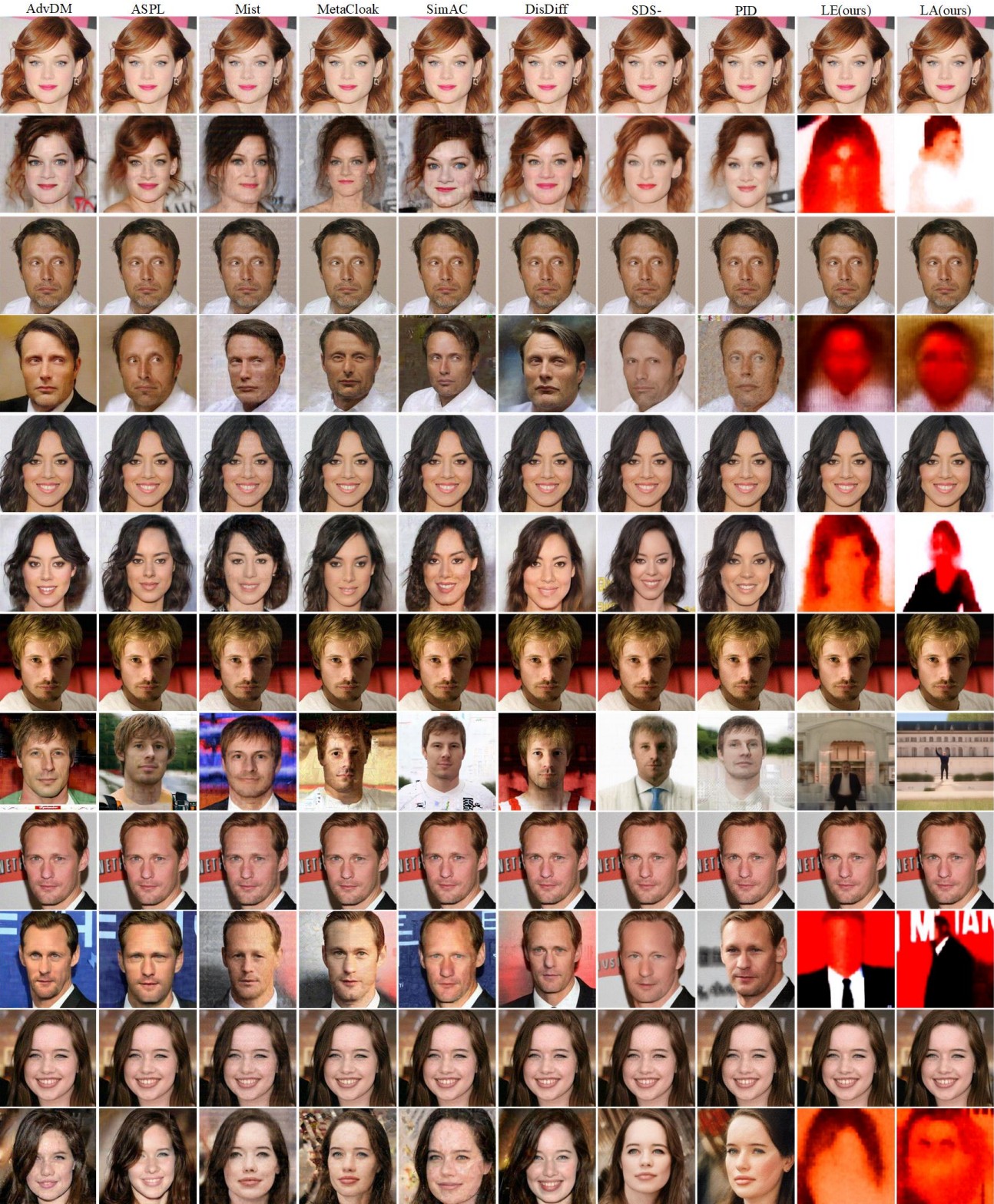

*Figure 9.* More visualization of our methods and baselines against **LoRA** on CelebA-HQ. The odd-numbered rows show reference images, while the even-numbered rows display generated images. Adversarial examples generated on SD1.5 are transferred to **SD3.5 Medium** as the training set. The noise originally added by the baselines is largely removed, indicating poor transferability. In contrast, our method still generates images that are unrelated to identity semantics, demonstrating better transferability. Best viewd in zoom.

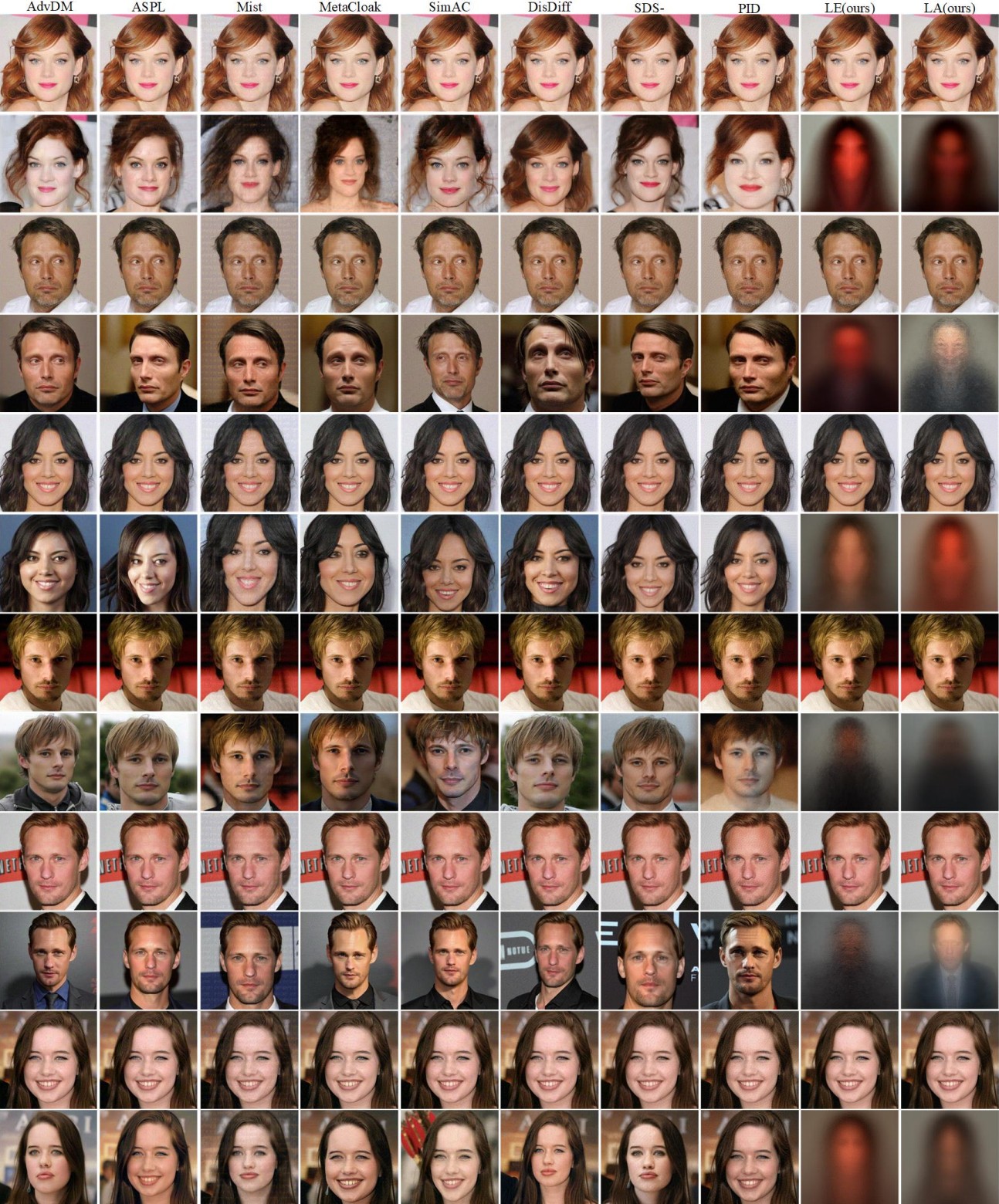

*Figure 10.* More visualization of our methods and baselines against **LoRA** on CelebA-HQ. The odd-numbered rows show reference images, while the even-numbered rows display generated images. Adversarial examples generated on SD1.5 are transferred to **FLUX.1-dev** as the training set. The noise originally added by the baselines is largely removed, indicating poor transferability. In contrast, our method still generates images that are unrelated to identity semantics, demonstrating better transferability. Best viewd in zoom.

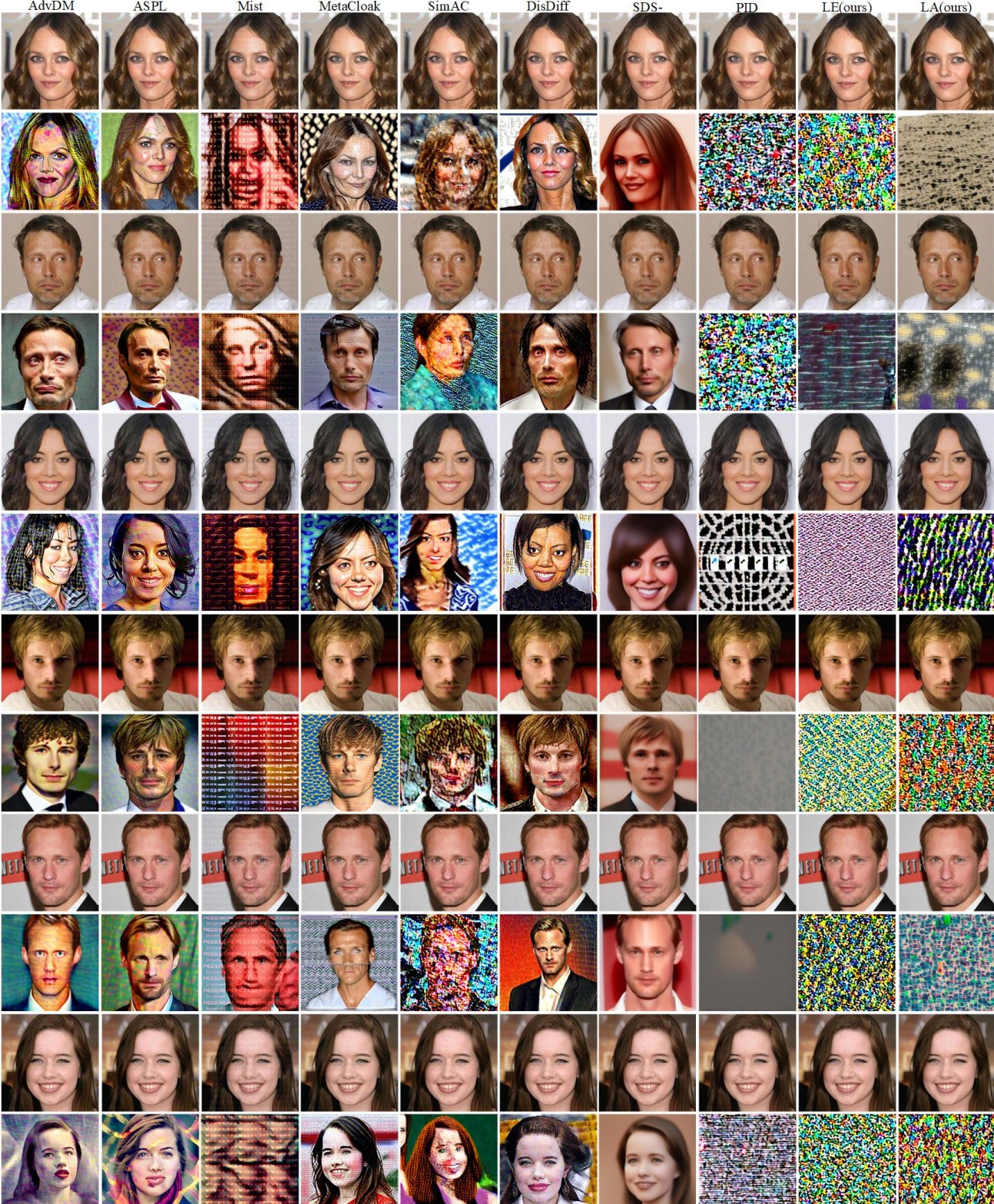

*Figure 11.* More visualization of our methods and baselines against **Textual Inversion** on CelebA-HQ. The odd-numbered rows show reference images, while the even-numbered rows display generated images. All baselines add different levels of noise to the generated images, while only our methods completely remove identity semantics and generate pure noise images.

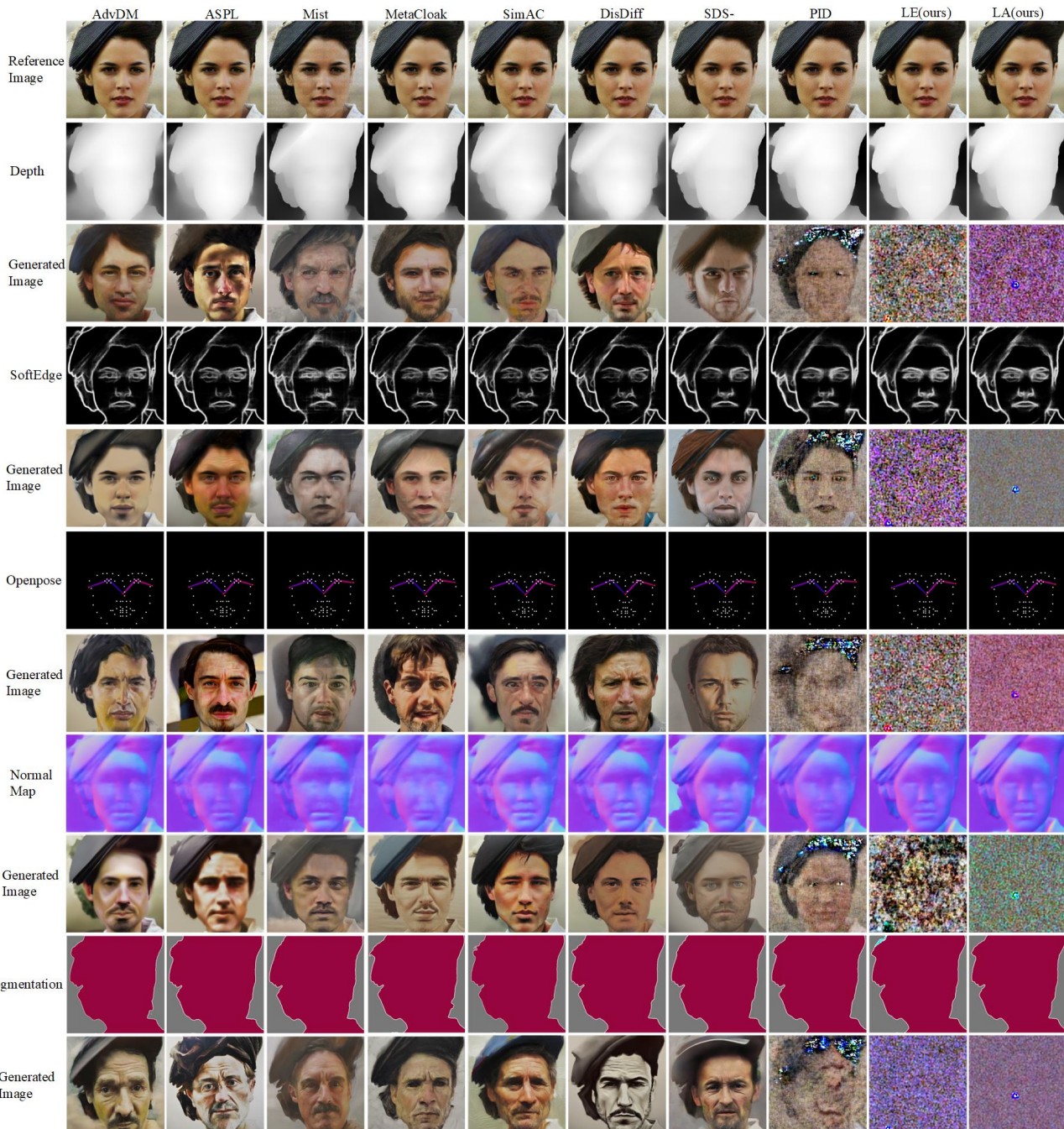

*Figure 12.* Visualization of our methods and baselines against **ControlNet-based Image Editing** ([Zhang et al., 2023](#)) on CelebA-HQ. We evaluate five image editing applications: Depth Map, SoftEdge, Openpose, Normal Map, and Segmentation. The prompt used is *"a man"*. All baselines add different levels of noise to the generated images, while only our method completely removes identity semantics and generates pure noise images.

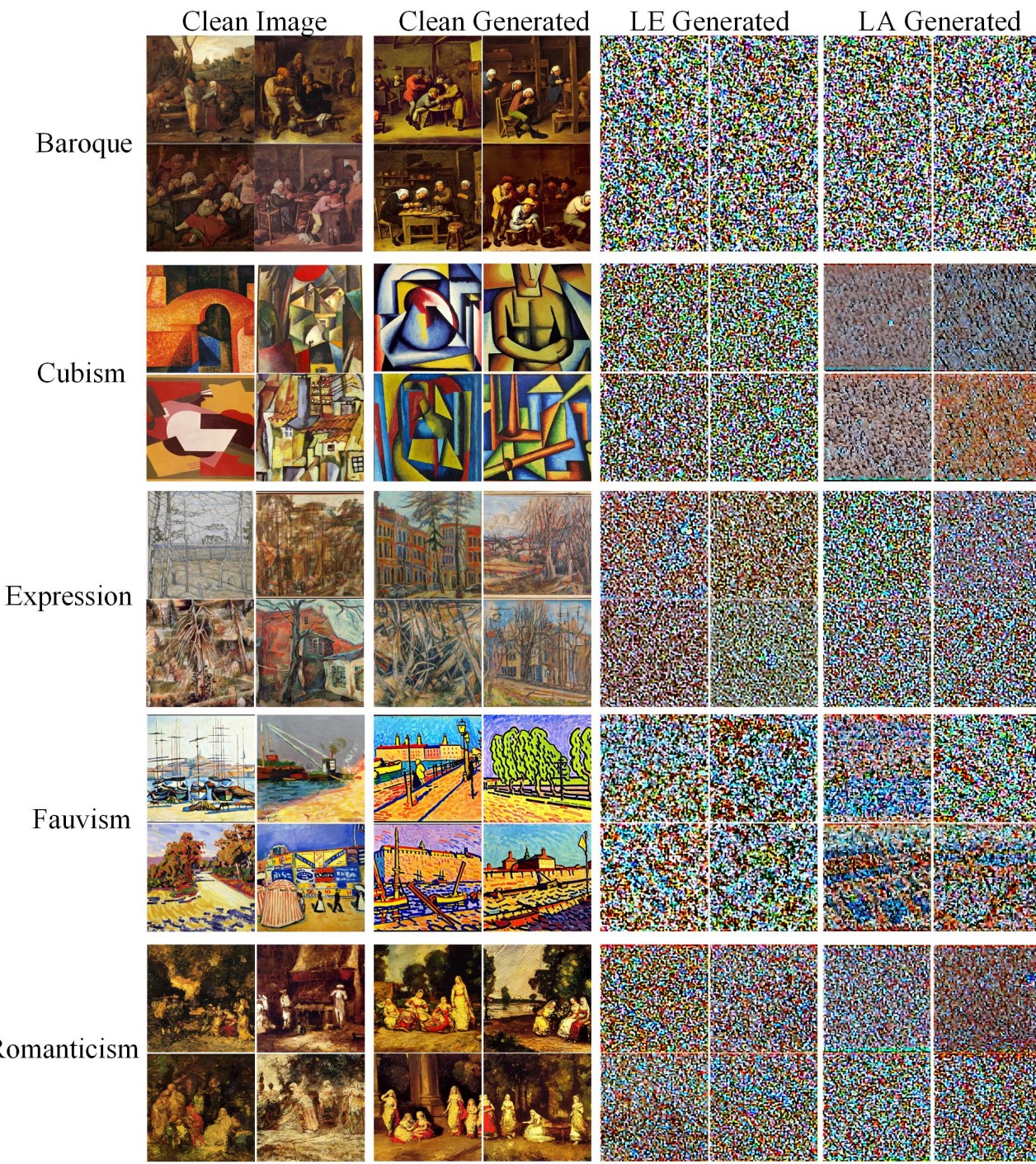

*Figure 13.* Visualization of our methods and baselines against **DreamBooth** on WikiArt. We experiment with five different artistic styles: Baroque, Cubism, Expressionism, Fauvism, and Romanticism. Our method consistently generates pure noise images, effectively protecting artistic styles from unauthorized imitation.

