# OpenReview forum: "Variance as a Catalyst: Efficient and Transferable Semantic Erasure Adversarial Attack for Customized Diffusion Models"
_ICML.cc/2025/Conference — ICML 2025 poster_

### Official Review · Reviewer_vD1g · 2025-03-02

**Overall Recommendation:** 3

**Summary:**

The paper proposes a novel adversarial attack method, leveraging variance manipulation to efficiently and consistently erase identity semantics from images generated by diffusion models, such as Stable Diffusion. The authors introduce two main approaches, Laplace-based and Lagrange Entropy, to address limitations in existing methods like MSE and PID . Their techniques focus on optimizing the variance components of the latent space, enabling more effective and efficient semantic erasure in generated images. Additionally, they demonstrate the transferability of their methods across various diffusion models and against different personalization techniques, showcasing the robustness of their approach.

**Claims And Evidence:**

Yes

**Essential References Not Discussed:**

No

**Experimental Designs Or Analyses:**

The experiments are well-designed, covering a range of datasets and comparing the proposed methods to a variety of baselines. However, the study could benefit from a broader set of experiments that test the methods on more diverse datasets, including non-human subjects or artistic images, to assess the method’s generalizability. Additionally, more discussion is needed on the visual quality of generated images and how the methods balance identity erasure with image naturalness.

**Methods And Evaluation Criteria:**

The paper proposes two novel loss functions—LA and LE—focused on optimizing the variance in the latent space of diffusion models. The LA loss ensures the gradient is aligned with the variance growth direction, allowing for efficient local optimization. The LE loss integrates entropy and a Lagrange constraint to balance optimization, promoting variance growth in smaller components and preventing slow convergence as variance becomes more uniform. These two losses outperform traditional methods like MSE and PID in both speed and effectiveness.

**Other Comments Or Suggestions:**

No

**Other Strengths And Weaknesses:**

# Strength

- The idea of manipulating variance components in latent space to achieve semantic erasure is an innovative approach. It provides an alternative to traditional adversarial techniques that focus on perturbing pixels or the mean of the latent space. This allows for more targeted attacks on identity information.
- The methods proposed (LA and LE) are computationally efficient, using significantly less memory and processing time compared to existing methods like MetaCloak and SimAC. For example, LE requires only 8 seconds and 4.5 GB of GPU memory, which is impressive considering the task at hand. Additionally, the high transferability of the methods across models like SD1.5, SD2.1, SDXL, and others is a valuable contribution to the fieldxperimental Design and Evaluation

# Weakness
- The mathematical models in Section 3 offer a detailed explanation of the loss functions (LA and LE), but the implications of these formulations on the stability of the optimization process, especially under high-dimensional settings, could be better discussed . While the authors addrestical challenges (e.g., Jacobian computation), more clarity on potential issues like gradient vanishing/explosion in high-dimensional latent spaces could enhance the robustness of their claims .
- The authors present a variety of experimental settings, including comparisons with multiple state-of-the-art methods. However, the experiments could benefit from a deeper analysis of the real-world implications of their method. For example, further discussion is needed on the trade-offs between the extent of identity erasure and the visual quality of generated images. Although the images achieve high semantic erasure, they may not always meet the quality standards expected in all applications.
- The experimental evaluation also primarily focuses on face images (CelebA-HQ, VGGFace2). It would be beneficial to extend these tests to a broader range of image types and tasks (e.g., artistic generation or non-human subjects) to test the generalizability of the proposed method.

**Questions For Authors:**

No

**Relation To Broader Scientific Literature:**

The paper is well-positioned in the context of adversarial attacks and privacy protection for generative models.

**Theoretical Claims:**

The paper claims that manipulating the variance of the latent codes is the key to erasing identity semantics in generated images. While the authors theoretically justify their approach using gradient flow analysis, the detailed reasons why variance manipulation is superior to other adversarial strategies could be elaborated further.

---

> ### Author Rebuttal · Authors · 2025-03-29
>
> ### **Q1: Gradient Explosion and Unstable Oscillations**
>
> Thank you for raising this important point.  We acknowledge that rapidly increasing latent variance can lead to gradient explosion and numerical instability. As discussed in **Appendix D.1 and Fig. 4**, the gradient of the LE loss, $\tfrac{\partial \mathcal{L}_{LE}}{\partial \delta}$, explodes around 30 optimization steps, while the LA gradient begins oscillating around step 50. These issues are caused by the nonlinearity of the model and the Jacobian terms, which become unstable as the latent distribution expands and certain activations saturate in high-dimensional space.
>
> Interestingly, **this instability is not entirely detrimental**. In fact, it reveals a clear and interpretable progression in the generated images over the course of optimization:
>
> -  **At step 20**, the variance increases slightly. Most latent dimensions remain compact, and only a few begin to expand. The generated image still retains a recognizable facial structure with mild noise and texture distortions.
> -  **Around step 25**, variance expansion becomes more pronounced. The image quality starts to degrade  with strong noise artifacts appearing across the face. However, some identity-related features are still visible. At this point, the visual result looks similar to outputs from methods like SimAC and PID—unnatural and distorted, but not fully erased.
> -  **Between steps 30 and 50**, the variance increases rapidly across many dimensions, causing the latent distribution to flatten and spread widely. This pushes the sampled latent codes far from the dense semantic region covered by the VAE’s training distribution. Since these codes fall outside the decoder’s learned prior, they are interpreted as random noise, leading to outputs with no recognizable facial structure or semantic content.
> -  **At step 75**,  variance grows at different rates across dimensions. Some dimensions saturate while others continue expanding, leading to imbalanced latent structures. The outputs exhibit chaotic combinations of textures and colors, forming a new kind of “random state” in appearance.
> -  **Beyond step 100**, further gradient explosion or sign flipping may occur. In some cases, the decoder maps these extreme latent codes to scene-like artifacts (e.g., indoor layouts). These are not restored content but hallucinated patterns caused by nonlinear decoding from noise.
>
> This behavior highlights a key advantage of our method: **Controllability**. Users can stop the optimization early (e.g., at step 25–30) to avoid instability while still achieving strong identity removal with reasonable image quality. Alternatively, further optimization gradually strengthens the erasure effect, progressing from facial distortion to complete identity removal and, eventually, to synthetic scene-like hallucinations. This allows users to tailor the erasure strength to their specific needs, offering a practical balance between robustness and flexibility.
>
>
>
> ### **Q2. Trade-off Between Identity Erasure and Visual Quality**
>
> Thank you for the thoughtful suggestion. We address the trade-off between identity erasure and visual quality in **Table 7 of the main paper**, where we compare our method under different perturbation budgets. Notably, our approach achieves complete identity removal (ISM = 0) even with a small perturbation of 8/255, while baseline methods require significantly larger perturbations (e.g., 0.05) to achieve similar or weaker results. This demonstrates that our method is not only effective but also visually practical.
>
> In addition, a key strength of our approach is its controllability. Users can flexibly adjust the perturbation size and the number of optimization steps to balance protection strength and image quality. For instance, stopping at around 30 steps with 8/255 perturbation yields strong identity erasure while maintaining acceptable visual quality, which we believe makes the method more adaptable for real-world applications.
>
>
>
> ### **Q3: More Experiments on other datasets**
>
> Thank you for this valuable suggestion. **As shown in Fig. 13 at the end of the appendix**, we have extended our evaluation to artistic-style images. Our method remains highly effective in this setting, producing pure noise outputs and demonstrating strong semantic removal. These results indicate that our approach is not limited to facial datasets and can generalize well to a broader range of image types. We sincerely appreciate your encouragement to further validate the robustness of our method.

---

### Official Review · Reviewer_36nD · 2025-03-04

**Overall Recommendation:** 3

**Summary:**

This paper protects images from malicious editing by attacking diffusion models. The authors design two loss functions, LA and LE, to attack the image variance after VAE encoding, demonstrating stronger attack effectiveness compared to other methods.

## update after rebuttal
Authors' rebuttal have solved my concerns, and I will adjust my rating based on other reviewers' scores.

**Claims And Evidence:**

Although the authors emphasize the importance of variance, there is no experiment demonstrating whether attacking the mean also has the same attack effect.

**Essential References Not Discussed:**

No

**Experimental Designs Or Analyses:**

The relationship between the two proposed losses is not clearly illustrated, and the experiments do not explicitly clarify which loss is more advantageous for specific scenarios.

**Methods And Evaluation Criteria:**

What does “clean” in Table 8 mean? Does it refer to the performance without any defense method applied, or the performance without any attack? Both should be included in the table to better illustrate the impact of the defense methods on the attacks.

**Other Comments Or Suggestions:**

No additional comments

**Other Strengths And Weaknesses:**

Strengths :
1. The proposed methods show better attack performance compared to other approaches.
2. The design of the loss functions has a certain level of interpretability.

Weakness see the above

**Questions For Authors:**

No additional questions.

**Relation To Broader Scientific Literature:**

The proposed method is an improvement based on previous VAE methods, such as PID.

**Theoretical Claims:**

In my opinion, there are no major issues with the theoretical analysis, but I am not an expert in this area, so I suggest considering the opinions of other reviewers as well.

---

> ### Author Rebuttal · Authors · 2025-03-29
>
> **Q1: Experiments of Mean Attack**
>
> **Table 1: Effectiveness of Attacking Mean vs. Variance**
>
> | Method| ISM ↓ | FDFR ↑ | Brisque ↑| LPIPS ↑|
> | -------------------- | ----- | ------ | ----------- | --------- |
> | LA_Mean_30step| 0.276 | 0.598  | 29.801| 0.855|
> | LA_Mean_50step| 0.234 | 0.703  | 31.714| 0.861|
> | LA_Mean_100step| 0.204 | 0.781  | 33.219| 0.871|
> | LA_Mix_100step| 0.231 | 0.772| 32.594| 0.872|
> | LA_Var_30step (ours) | **0** | **1**  | **155.845** | **0.959** |
>
> Thank you for your thoughtful comment. We address this concern both theoretically (see our response to Reviewer WD7W, Q1) and empirically in **Table 1**. The results clearly show that attacking the mean alone yields significantly weaker erasure performance, even with more optimization steps. For example, **LA_Mean_100step** achieves ISM = 0.204, whereas our **LA_Var_30step** achieves **ISM = 0** and **FDFR = 1** with much fewer steps. Furthermore, combining mean and variance attacks (**LA_Mix_100step**) limits the growth of variance and results in less effective erasure than attacking variance alone. These consistent trends demonstrate that variance-based attacks are more effective and efficient, both in theory and in practice.
>
>  **Q2: Ambiguity about Table 8**
>
> **Table 2: Robustness of different methods against JPEG Compression and GrIDPure**
>
> | Attack Method | No Image Preprocessing | No Image Preprocessing | No Image Preprocessing | No Image Preprocessing | JPEG Compression | JPEG Compression | JPEG Compression | JPEG Compression | GrIDPure| GrIDPure| GrIDPure   | GrIDPure  |
> | ------------- | ---------------------- | ---------------------- | :--------------------: | ---------------------- | ---------------- | ---------------- | ---------------- | ---------------- | --------- | --------- | ---------- | --------- |
> || ISM ↓| FDFR ↑|Brisque ↑| LPIPS ↑| ISM ↓| FDFR ↑| Brisque ↑| LPIPS ↑| ISM ↓| FDFR ↑| Brisque ↑  | LPIPS ↑|
> | Clean Image   | 0.608| 0.041|17.896| 0.662| -| -| -| -| -| -| -| -|
> | AdvDM| 0.424| 0.307|24.215| 0.798| 0.659| 0.031| 30.269| 0.771| 0.601| 0.031| 20.609| 0.705|
> | ASPL| 0.406| 0.287|24.419| 0.805| 0.668| 0.042| 31.592| 0.769| 0.574| 0.022| 20.061| 0.707|
> | Mist| 0.249| 0.169|13.981| 0.707| 0.629| 0.014| 33.686| 0.807| 0.541| 0.057| 20.149| 0.773|
> | MetaCloak| 0.593| 0.051|36.325| 0.712| 0.592| 0.075| 53.622| 0.813| 0.593| 0.047| 27.871| 0.745|
> | SimAC| 0.253| 0.865|51.059| 0.823| 0.632| 0.032| 32.955| 0.759| 0.548| 0.091| 24.967| 0.699|
> | DisDiff| 0.605| 0.116|29.361| 0.695| 0.672| 0.035| 29.217| 0.753| 0.618| 0.032| 19.651| 0.693|
> | SDS-| 0.655| 0.005|38.519| 0.743| 0.684| 0.022| 33.053| 0.818| 0.624| 0.047| 23.174| 0.729|
> | PID| 0.069| 0.938|85.533| 0.899| 0.473| 0.255| 40.435| 0.804| 0.473| 0.189| 43.878| **0.773** |
> | **LE (ours)** | **0**| **1**|**155.804**| **0.947**| **0.451**| **0.358**| **62.541**| **0.821**| **0.456** | **0.231** | **49.071** | 0.769     |
> | **LA (ours)** | **0**| **1**|**155.845**| **0.959**| **0.447**| **0.342**| **63.572**| 0.791| **0.449** | **0.242** | **49.729** | 0.761     |
>
> Thank you for the question, and we apologize for the confusion. In Table 8, “Clean” refers to the original image without adversarial perturbation.  To better illustrate the robustness of our method, we provide **Table 2**, which evaluates robustness performance under both **traditional image preprocessing (JPEG Compression)** and **diffusion-based purification (GrIDPure)**.
>
> In this table:
>
> - “Clean Image” denotes the unperturbed original image
> -  “No Image Preprocessing” indicates adversarially protected images before any additional post-processing.
>
> As shown, our methods consistently achieve stronger robustness than most baselines and deliver performance comparable to or better than PID under both JPEG compression and GrIDPure purification. This demonstrates the effectiveness and reliability of our approaches under common real-world applications.
>
>  **Q3：LA vs LE, which is better?**
>
> Thank you for the feedback. Both LA and LE losses are designed to align perturbations with the direction of latent variance growth, and they share the same core objective. While their formulations differ slightly, they are conceptually equivalent in guiding effective attacks.
>
> As shown in **Tables 1, 2, and 3 of the main paper**, both losses consistently outperform existing baselines across different evaluation settings. Table 4 further demonstrates that our method improves attack efficiency by nearly 30× compared to the previous SOTA method PID. In addition, Table 7 shows that our loss functions achieve complete identity erasure (e.g., ISM = 0, Brisque = 122.266) even with a small perturbation budget of 8/255, while baselines require much larger perturbations (e.g., 0.05) to achieve inferior results.
>
> These results suggest that both losses are effective and efficient, and users can choose either based on implementation preference or training stability.

---

### Official Review · Reviewer_93PG · 2025-03-13

**Overall Recommendation:** 4

**Summary:**

This paper proposes two novel loss functions, i.e., Laplace Loss (LA) and Lagrange Entropy Loss (LE), which used for adversarial attacks aimed at disrupting Latent Diffusion Models (LDMs). The key insight is identifying the variance of the VAE latent code as critical for effectively erasing identity semantics in generated images. Experimental results show that the proposed methods achieve good performance compared to existing techniques, effectively producing pure-noise images with completely erased identity semantics. Additionally, these methods demonstrate better transferability and reduced computational requirements.

**Claims And Evidence:**

The claims made regarding the efficiency and efficacy of the proposed loss functions (LA and LE) are supported by experimental evidence. State-of-the-art performance metrics (ISM, FDFR, Brisque, LPIPS) demonstrate strong identity erasure capabilities compared to baseline solutions. However, some minor issues remain:

[Issue 1] Clarification of the attack scenario is required. It is suggested to clearly state the training and testing process since not all the readers understand the process. For example, the customization of DM usually require the binding of a specific token to the concept.
[Issue 2] Evaluation of the protected image quality beyond distortion metrics could strengthen evidence supporting the practical usability of the proposed methods.

**Essential References Not Discussed:**

The related works section appears comprehensive.

**Experimental Designs Or Analyses:**

The experimental designs are sound, clearly comparing proposed methods to multiple baselines. However, the design is limited by testing only one type of prompt. It is suggested to test different kinds of prompts, as well as the prompts that will be used in practical scenarios.

**Methods And Evaluation Criteria:**

The methods proposed are appropriate for the stated goal of semantic erasure in generative diffusion models. However, the choice of metrics differs from related works without explicit justification.

**Other Comments Or Suggestions:**

LINE 196: Replace "by another neural network" with "by another neural network $g$".
Table 6: Alignment issues with check and cross symbols should be addressed for clarity.

**Other Strengths And Weaknesses:**

**Strengths**:

S1. Clear Theoretical Justification: The proposed novel loss functions (Laplace-based Loss and Lagrange Entropy-based Loss) are deeply grounded in theoretical analysis, providing a robust rationale for variance manipulation as a mechanism for semantic erasure.

S2. Computational Efficiency: The methods achieve substantial speed improvements and require significantly fewer computational resources, making them practical even in resource-constrained settings.

S3. Empirical Performance: The experiments demonstrate state-of-the-art performance across several datasets and diffusion model architectures, including advanced versions such as SD3.5 and FLUX.1-dev, validating the effectiveness and generalizability of the approach.

(4) High Transferability: Unlike prior methods, the proposed techniques demonstrate consistently high transferability across diverse diffusion model architectures, increasing their utility in various real-world scenarios.

Weaknesses:

W1. Limited Variety in Prompt Selection: The experimental evaluations rely heavily on a single prompt structure (e.g., "a photo of a sks person"), restricting the assessment of the generalizability of the attack. Expanding tests to diverse prompt types would better demonstrate the broader applicability of the proposed approach, for example, a photo of a sks person dancing/eating on the train.

W2. Introduction of New Metrics Without Comprehensive Justification: The authors introduce new evaluation metrics, such as Identity Score Matching (ISM) and Face Detection Failure Rate (FDFR), without thoroughly justifying why established metrics (e.g., standard face recognition scores or perceptual quality measures) were insufficient. This lack of detailed explanation may limit the perceived validity and broader acceptance of the evaluation criteria.

W3. The results of adversarial training are encouraged.

**Questions For Authors:**

Q1. Can you provide a more explicit clarification regarding why only a single type of prompt was used, and discuss potential impacts on the generalization and robustness of your approach?

Q2. Could you elaborate on the rationale for adopting ISM, FDFR, and LPIPS as evaluation metrics over previously established metrics like FDS, FID, and IQS?

Q3. What if we do the adversarial training on VAE?

**Relation To Broader Scientific Literature:**

This paper posits its contributions within the context of existing literature, highlighting improvements over current methods in adversarial attacks for LDMs. It effectively identifies gaps in previous approaches (inefficient optimization, misalignment of gradient signs) and clearly differentiates its contributions.

**Theoretical Claims:**

The paper provides thorough theoretical insights into the proposed loss functions, and the derivations appear sound upon careful inspection.

---

> ### Author Rebuttal · Authors · 2025-03-29
>
> ### **Q1: Attack and Defense Scenario**
>
> Thank you for raising this point.  Our method is designed for a practical adversarial setting involving a victim (User A) and an attacker (User B):
>
> - **Defense Phase**: User A wishes to share photos online but wants to prevent misuse by personalization techniques like DreamBooth or LoRA. Before uploading, User A applies our protection method to the images, which introduces subtle perturbations on photos.
>
>
> - **Attack Phase**: User B collects these protected images and tries to train a personalized diffusion model to generate fake content involving User A. However, since the images have been preemptively protected, the model fails to capture meaningful identity information and generates pure noise output.
>
> ### **Q2: Trade-off Between Identity Erasure and Visual Quality**
>
> Thank you for the thoughtful comment. We address this trade-off in Table 7 of the main paper, where we compare performance under different perturbation budgets. Our method achieves complete identity erasure (e.g., ISM = 0) even with a small perturbation of 8/255, outperforming baselines that require much larger distortions (e.g., 0.05) to reach comparable effectiveness. This indicates that our method is both effective and visually practical.
>
> In real-world deployment, users can easily balance protection strength and visual quality by adjusting the perturbation budget and number of optimization steps. For instance, limiting the perturbation to 8/255 and stopping at 30 steps provides strong identity removal while preserving reasonable image quality.
>
> ### **Q3: Different Prompt**
>
> Thank you for raising  this important concern. As shown in **Appendix C.1 (Table 9)** and **Appendix E (Figure 6)**, we evaluate our method using an alternative prompt (“**a dslr portrait of sks person**”) in a DreamBooth personalization setting. The results show that our approach remains highly effective under prompt mismatch, consistently producing pure-noise outputs. This confirms that our method is **prompt-agnostic**.
>
> Unlike prior works such as Anti-DreamBooth, AdvDM, SimAC, and MetaCloak, which require prompt-specific gradient information during optimization, our method does not bind perturbations to any particular prompt. Instead, it directly targets the VAE encoder, which operates before the prompt is introduced. Since prompts influence only the denoising stage (e.g., U-Net or MM-DiT), they do not affect our gradient path, ensuring independence from specific prompt conditions.
>
> Moreover, our method demonstrates strong performance against **LoRA-based personalization (Appendix E, Figure 7)**, where arbitrary prompts can be used during generation, as well as in **ControlNet-based image editing (Appendix E, Figure 28)**, where we apply a new prompt (“**a man**”) to modify the gender of the reference image. These results further validate the prompt-agnostic nature of our approach and highlight its robustness and practicality in diverse real-world scenarios.
>
> ### **Q4: Metric**
>
> Thank you for the question. Our choice of ISM, FDFR, and BRISQUE, aligns with Anti-DreamBooth, a seminal work in identity erasure. These metrics provide intuitive and reliable evaluations for our task:
>
> - FDFR (Face Detection Failure Rate) measures whether a face is undetectable and is mathematically equivalent to $1-FDS$. We report FDFR instead of FDS because it more intuitively reflects successful cases of identity erasure in privacy protection.
> - ISM evaluates identity similarity between the original and generated images using a face recognition model, directly reflecting whether identity semantics remain.
> - LPIPS is  a perceptual metric to estimate visual degradation. Compared to traditional metrics such as FID or IQS, LPIPS better correlates with human perception, especially in assessing localized texture changes or structural distortions.
>
> ### **Q5: Adversarial Training on VAE**
>
> Adversarial training on the VAE may theoretically enhance robustness, but it faces two key challenges:
>
> **1. High Cost and Training Difficulty**
>  Adversarial training is computationally expensive and time-consuming. This is especially problematic for VAEs, which are harder to train than classifiers due to issues like posterior collapse and poor convergence with limited or adversarial data. For typical users with limited resources, this makes adversarial training inefficient and impractical.
>
> **2. Stronger Attacks via Proxy Optimization**
>  As shown in ASPL, alternating optimization using the VAE as a proxy can significantly boost attack strength. This increases the difficulty of robust training and further destabilizes the learning process.
>
> These two issues suggest that while adversarial training is a theoretically viable defense, it is both computationally costly and technically nontrivial in practice, especially in low-resource environments.

---

### Official Review · Reviewer_WD7W · 2025-03-14

**Overall Recommendation:** 3

**Summary:**

The paper introduces LA and LE loss functions to enhance semantic erasure in customized diffusion models, addressing privacy concerns by completely removing identity-related features. It identifies variance in VAE latent codes as key to image distortion and uses optimized variance expansion for effective erasure. Experiments on CelebA-HQ and VGGFace2 show state-of-the-art identity removal, 30x faster optimization, and strong transferability across diffusion models. The method is computationally efficient.

**Claims And Evidence:**

Yes

**Essential References Not Discussed:**

N/A

**Experimental Designs Or Analyses:**

Yes, I have reviewed the soundness and validity of the experimental designs and analyses.

**Methods And Evaluation Criteria:**

The proposed LA and LE loss functions effectively address the semantic erasure problem in diffusion models by leveraging variance expansion in VAE latent codes. The methodology is well-grounded in theoretical analysis and overcomes limitations of prior approaches. The evaluation criteria are appropriate, using CelebA-HQ and VGGFace2 datasets, and metrics such as Identity Score Matching (ISM), Face Detection Failure Rate (FDFR), Brisque, and LPIPS, which provide a comprehensive assessment of identity removal effectiveness.

**Other Comments Or Suggestions:**

N/A

**Other Strengths And Weaknesses:**

N/A

**Questions For Authors:**

1. Existing methods may not be optimal, but they seem sufficient to make people doubt the authenticity of a generated image. For instance, in Figure 3, results from SimAC and PID are not as effective as those from your proposed method, but they still make the image look unnatural enough for a viewer to recognize it as manipulated. Could you elaborate on your argument regarding why stronger erasure is necessary beyond this level of distortion?
2. Given that your proposed methods rapidly expand variance, is there a potential risk of gradient instability or numerical issues during optimization?
3. Your method demonstrates efficiency in terms of GPU usage and runtime. Could you comment on how this computational efficiency scales to larger images or video data, given the rapidly increasing usage of diffusion models in these domains?
4. While your method shows strong transferability across different model architectures, could you discuss any theoretical insights explaining why variance-based attacks generalize so effectively across diverse diffusion models?

**Relation To Broader Scientific Literature:**

This paper builds on prior work in adversarial attacks on diffusion models and semantic erasure techniques, particularly targeting identity removal in personalized generative models. Previous methods, such as PID and Mist, attempted to erase identity semantics but were limited by slow convergence, poor transferability, and high computational costs.

**Theoretical Claims:**

Please refer previous section.

---

> ### Author Rebuttal · Authors · 2025-03-29
>
> **Q1: Advantages of Variance-based Attack and Better Transferability**
>
>  **1.Model Architecture**
>
> Earlier diffusion models (e.g., SD1.5, SD2.1) use U-Net backbones, enabling effective attacks based on U-Net gradients or cross-attention. However, newer models like SD3.5 and FLUX.1 adopt Transformer-based backbones such as MM-DiT, which differ greatly in gradient behaviors. As a result, prior gradient-based attacks often fail. Despite this, all Latent Diffusion Models share a common pipeline: a VAE encodes the input image to a latent space, followed by diffusion. Our method perturbs the input image to corrupt the VAE output, an essential bottleneck, ensuring transferability across different architectures.
>
>  **2.Attack VAE Mean $\mu$**
>
> VAE applies the reparameterization trick to make sampling differentiable:
>
> $$
> z=\mu(x)+\sqrt{\sigma^2(x)}\odot \epsilon,\epsilon \sim \mathcal{N}(0,I).
> $$
> Perturbing the mean yields:
>
> $$
> z'=\mu(x)+\delta_\mu+\sqrt{\sigma^2(x)}\odot\epsilon, \epsilon \sim \mathcal{N}(0,I).
> $$
> The corresponding probability density function (PDF) becomes:
>
> $$
> p(z')=\frac{1}{\sqrt{2\pi \sigma^2(x)}}exp(-\frac{(z'-(\mu(x)+\delta_\mu))^2}{2\sigma^2(x)}).
> $$
> Attacking the mean shifts the center of the latent distribution without altering the shape of its PDF. Since the variance remains unchanged, the distribution stays compact. Due to the translation-invariance of convolutional backbones, the semantic structure is partly preserved. Visually, this effect is mostly limited to local texture distortion, appearing as noise artifacts rather than semantic erasure.
>
> **3.Attack VAE Variance $\sigma^2$:**
>
> Perturbing the variance gives:
> $$
> z=\mu(x)+\sqrt{\sigma^2(x)+\delta_{\sigma^2}}\odot \epsilon,\epsilon \sim \mathcal{N}(0,I)
> $$
> The PDF becomes:
> $$
> p(z')=\frac{1}{\sqrt{2\pi(\sigma^2(x)+\delta_{\sigma^2}})}exp(-\frac{(z'-\mu(x)^2)}{2(\sigma^2(x)+\delta_\sigma^2)}).
> $$
> Here, **$\mu(x)$ represents the mean of the original latent distribution**, which corresponds to the dense, semantically meaningful region learned during training.
>
> As variance grows large, the 1D Gaussian distribution of each latent dimension flattens into a wide and low curve that is nearly parallel to the x-axis. This flattening means that the probability density near the original  mean $\mu (x)$ drops quickly, while the overall sampling interval expands considerably. As a result, the latent samples are no longer concentrated near the dense semantic center of the distribution but become spread out over a vast latent space.
>
> From a high-dimensional perspective, this effect is amplified. Given a $d$-dimensional latent space, the expected squared distance between the latent samples $z'$ and mean  $\mu (x)$ is:
> $$
> \mathbb{E}[\|| z' - \mu(x) \||^2] = \sum_{i=1}^d \left( \sigma_i^2(x) + \delta_{\sigma^2,i} \right)
> $$
> As variance grows, latent samples $z'$ are pushed away from the high-density region centered around $\mu(x)$. These samples fall outside the data manifold and become unrecognizable to the decoder's prior knowledge. Consequently, the decoder fails to map them to any meaningful structure and instead produces pure noise images.
>
> **Q2: Why Stronger Erasure is Necessary**
>
> Existing methods add noise artifacts that make the image appear unnatural, but still preserve recognizable facial structure. Such noise artifacts added on preserved facial structures often induce a sense of visual eeriness and discomfort. Moreover, this kind of incomplete protection can also be seen as a form of visual uglification of the victim’s appearance, which could potentially be exploited by malicious users.  In contrast, our method aims for complete semantic erasure, providing stronger privacy protection.
>
>  **Q3: Higher Resolution and Video tasks**
>
> Thank you for the insightful question. We conducted tests with higher-resolution inputs (e.g., 1024×1024). When preserving the adversarial example at full 1024×1024 resolution, the runtime increases from approximately 8 seconds to 19 seconds.
>
> This is primarily due to the increased computational cost from larger spatial dimensions in both the VAE and the diffusion backbone, which affects all latent-space or diffusion-based attacks. Nevertheless, our method remains highly efficient and consistently outperforms existing approaches in both runtime and GPU usage, even at higher resolutions.
>
> For video data, our approach can be extended to video generation models using 3D-VAEs or spatiotemporal latent spaces. In this context, variance-based perturbations may disrupt temporal consistency or causal dependencies across frames, offering a promising direction for future adversarial research in video diffusion models.
>
>
>
> **Q4: Gradient Explosion and Unstable Oscillations**
>
> We appreciate your attention to this important point. Due to character limit, we provide a detailed response to a similar question raised by **Reviewer vD1g Q1**. We kindly invite you to refer to that reply for further clarification.

---

### Decision · Program_Chairs · 2025-05-01

**Decision:**

Accept (poster)

**Comment:**

The paper was reviewed by four experts in the field and finally received all positive scores: Weak accept, Accept, Weak accept, and Weak accept.
The major concerns of the reviewers are:
1.	why variance-based attacks are superior to other adversarial strategies,
2.	additional experimental results on other datasets and other settings.
The authors address the above concerns during the discussion period. Hence, I make the decision to accept the paper.